# Calibrated Self-Rewarding Vision Language Models

**Yiyang Zhou**[1]*, **Zhiyuan Fan**[5]*, **Dongjie Cheng**[6]*, **Sihan Yang**[7], **Zhaorun Chen**[2]
**Chenhang Cui**[8], **Xiyao Wang**[3], **Yun Li**[1], **Linjun Zhang**[4], **Huaxiu Yao**[1]
[1]UNC-Chapel Hill, [2]University of Chicago, [3]University of Maryland
[4]Rutgers University, [5]HKUST , [6]PolyU, [7]NTU, [8]NUS
yiyangai@cs.unc.edu, huaxiu@cs.unc.edu

## Abstract

Large Vision-Language Models (LVLMs) have made substantial progress by integrating pre-trained large language models (LLMs) and vision models through instruction tuning. Despite these advancements, LVLMs often exhibit the hallucination phenomenon, where generated text responses appear linguistically plausible but contradict the input image, indicating a misalignment between image and text pairs. This misalignment arises because the model tends to prioritize textual information over visual input, even when both the language model and visual representations are of high quality. Existing methods leverage additional models or human annotations to curate preference data and enhance modality alignment through preference optimization. These approaches are resource-intensive and may not effectively reflect the target LVLM's preferences, making the curated preferences easily distinguishable. Our work addresses these challenges by proposing the Calibrated Self-Rewarding (CSR) approach, which enables the model to self-improve by iteratively generating candidate responses, evaluating the reward for each response, and curating preference data for fine-tuning. In the reward modeling, we employ a step-wise strategy and incorporate visual constraints into the self-rewarding process to place greater emphasis on visual input. Empirical results demonstrate that CSR significantly enhances performance and reduces hallucinations across ten benchmarks and tasks, achieving substantial improvements over existing methods by 7.62%. Our empirical results are further supported by rigorous theoretical analysis, under mild assumptions, verifying the effectiveness of introducing visual constraints into the self-rewarding paradigm. Additionally, CSR shows compatibility with different vision-language models and the ability to incrementally improve performance through iterative fine-tuning. Our data and code are available at https://github.com/YiyangZhou/CSR.

## 1 Introduction

Large Vision-Language Models (LVLMs) [1–4] have achieved significant success by incorporating pre-trained large language models (LLMs) and vision models through instruction tuning. However, these LVLMs suffer from the hallucination phenomenon [5], which generates text responses that are linguistically plausible but contradict the visual information in the accompanying image. For instance, the description generated by LVLMs may include visual elements that are not depicted in the image. This issue can also occur when the LLM is highly factual and the visual backbone is capable of producing sufficiently high-quality representations. As indicated in Cui et al. [6], Guan et al. [7], the potential reason for this lies in the misalignment problem between image and text modalities in LVLMs, which causes the model to prioritize the text knowledge present in the training language data while ignoring the actual visual input information.

---

*Equal contribution

38th Conference on Neural Information Processing Systems (NeurIPS 2024).

Several works have been proposed to enhance modality alignment capability in LVLMs through preference fine-tuning techniques, such as reinforcement learning from human feedback (RLHF) [8] and direct preference optimization (DPO) [9, 10]. However, these methods often either introduce additional models, such as GPT-4, or depend on human annotation to generate preference data. This data generation process is not only resource-intensive but, more critically, fails to capture the inherent preferences of the target LVLM. Consequently, the target LVLM may easily discern preferences from such curated data, making them less effective (detailed analysis provided in Appendix A.4). Recently, self-rewarding approaches have emerged, utilizing a single LLM for both response generation and preference modeling, showing promising results in LLM alignment [11, 12]. Unlike LLMs, LVLMs face modality misalignment issues in both response generation and preference modeling stages, potentially resulting in self-generated preferences overlooking visual input information. Directly applying these self-rewarding approaches to LVLMs is not capable of addressing the modality alignment problem and redirecting LVLM's attention towards emphasizing input image information.

To tackle these challenges, our work introduces the **C**alibrated **S**elf-**R**ewarding (**CSR**) approach, aimed at calibrating the self-rewarding paradigm by incorporating visual constraints into the preference modeling process. Specifically, we train the target LVLM using an iterative preference optimization framework that continuously generates preferences and optimizes the target LVLM over multiple iterations. Starting with a seed model, each iteration employs sentence-level beam search [13, 14] to produce fine-grained candidate responses for each image and text prompt. During the beam search, for each generated sentence, we first utilize the language decoder to establish an initial reward (i.e., sentence-level cumulative probabilities). Subsequently, we calibrate this initial reward by incorporating an image-response relevance score, resulting in the calibrated reward score. These calibrated reward scores are utilized to guide the generation of the next batch of candidate sentences. Finally, responses with the highest and lowest cumulative calibrated reward scores are identified as preferred and dispreferred responses, respectively, for preference fine-tuning in the subsequent iteration.

The primary contribution of this paper is CSR, a novel calibrated self-rewarding paradigm for improving modality alignment in LVLMs. Theoretically, with mild assumptions, we show that introducing visual constraints in the self-rewarding paradigm can improve performance. Empirically, when compared with other competitive approaches (see Figure 1 for some representative methods), the results demonstrate that CSR is capable of improving performance on comprehensive LVLM evaluation benchmarks, VQA tasks, and reducing hallucination, achieving up to a 7.62% improvement on average. Additionally, we demonstrate CSR is capable of continuously improving performance over iterations, compatible with different large vision-language backbone models, and redirecting the attention of LVLMs toward the visual modality to achieve stronger modality alignment.

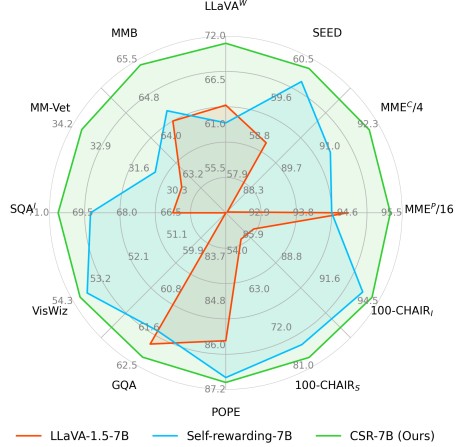

Figure 1: Benchmark performance comparison.

## 2 Preliminaries

In this section, we will provide a brief overview of LVLM and preference optimization.

**Large Vision Language Models.** LVLMs extend LLMs to multimodal scenario, which progressively predict the probability distribution of the next token for each input prompt. Given an <image $x_v$, text $x_t$> pair as input prompt $x$, LVLM outputs a text response $y$.

**Preference Optimization.** Preference optimization has shown promise in fine-tuning language models and aligning their behavior with desired outcomes. Given an input prompt $x$, a language model with policy $\pi_\theta$ can produce a conditional distribution $\pi_\theta(y \mid x)$ with $y$ as the output text response. The preference data is defined as $\mathcal{D} = \{(x^{(i)}, y_w^{(i)}, y_l^{(i)})\}_{i=1}^N$, where $y_w^{(i)}$ and $y_l^{(i)}$ denote the preferred

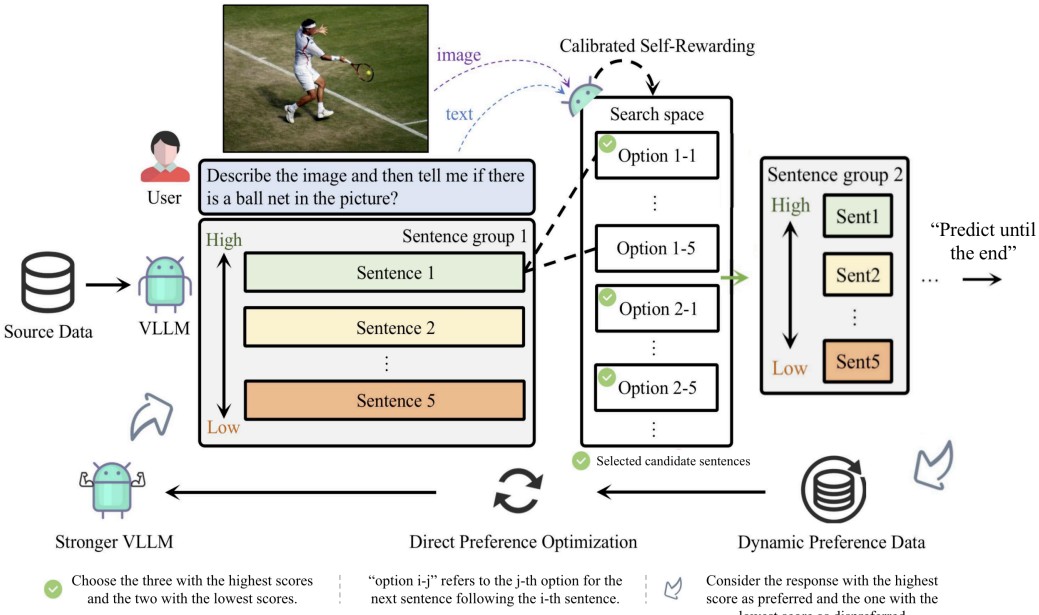

Figure 2: The CSR framework operates an iterative process of preference data generation and learning. During preference data generation, CSR utilizes a sentence-level beam search approach to construct responses sentence by sentence, assigning a reward to each sentence. This reward, initially generated by the model itself, is then calibrated using image-relevance information. Preferences are determined based on the cumulative reward for each response. In each iteration, CSR generates new preference data and performs preference learning based on this data, continuously enhancing the model's performance.

and dispreferred responses for the input prompt $x^{(i)}$. Preference optimization leverage the preference data to optimize language models. Taking DPO [15] as a representative example,, it formulates the probability of obtaining each preference pair as $p(y_w \succ y_l) = \sigma(r(x, y_w) - r(x, y_l))$, where $\sigma(\cdot)$ is the sigmoid function. DPO optimizes the language models with the following classification loss:

$$\mathcal{L}_{DPO}(\pi_\theta; \pi_{\text{ref}}) = -\mathbb{E}_{(x,y_w,y_l)\sim\mathcal{D}} \left[ \log \sigma \left( \alpha \log \frac{\pi_\theta(y_w|x)}{\pi_{\text{ref}}(y_w|x)} - \alpha \log \frac{\pi_\theta(y_l|x)}{\pi_{\text{ref}}(y_l|x)} \right) \right], \quad (1)$$

where $\pi_{\text{ref}}(y|x)$ represents the reference policy, i.e., the language model after performing supervised fine-tuning.

## 3 Calibrated Self-Rewarding Vision Language Models

To address this challenge, we propose **C**alibrated **S**elf-**R**ewarding (**CSR**), a novel approach aimed at improving modality alignment in LVLMs by integrating visual constraints into the self-rewarding paradigm. As illustrated in Figure 2, CSR trains the target LVLM by alternately performing two stages: candidate response generation and preference curation and fine-tuning. In the candidate response generation stage, we employ sentence-level beam search for each input prompt to produce fine-grained candidate responses. During this process, the language decoder determines the initial reward for each generated sentence, which is then calibrated by incorporating an image-response relevance score. This calibrated reward score guides the generation of subsequent sentences and finally generate the entire response. Moving on to the preference curation and fine-tuning stage, we use the responses with the highest and lowest cumulative calibrated rewards to construct the preferred and dispreferred responses, and utilize the constructed preference pairs for fine-tuning. In the remaining of this section, we will provide detailed explanations of CSR.

## 3.1 Step-Level Reward Modeling and Calibration

Before delving into how to generate candidate response and construct preference data, in this section, we first discuss how to formulate the reward within CSR. The ideal reward in the LVLM fulfills two specific criteria:

- Vision-Constrained Reward: This aspect aims to integrate image-relevance information into the reward definition of LVLMs. By doing so, we address the limitation of LVLM in overlooking image input data when generating preferences.

- Step-Wise Reward: Instead of assigning a single reward for the entire response, we opt for a step-wise approach. This involves assigning rewards at each step of response generation. Compared to a single reward, this finer-grained reward offers more detailed guidance and is more robust.

To fulfill these criteria, we propose a step-wise calibrated reward modeling strategy. Inspired by Process-Supervised Reward Models [16], we assign a reward score, $R(s)$, to each generated sentence $s$ during the sentence-level beam search. This score is a combination of two components: the self-generated instruction-following score, $R_T(s)$, and the image-response relevance score, $R_I(s)$.

Specifically, the self-generated instruction-following score, $R_T(s)$, is calculated using the language decoder of the LVLM. It represents the sentence-level cumulative probability of generating sentence $s$, formulated as:

$$R_T(s) = \prod_{t=1}^{N_o} P(r_o \mid x, r_1, r_2, \ldots, r_{o-1}), \tag{2}$$

where $N_o$ is the number of tokens in sentence $s$ and $r_o$ represents token $o$ in sentence $s$. A higher self-generated instruction-following score indicates a stronger capability of the generated response to follow instructions.

While the self-generated instruction-following score partially reflects the LVLM's preference, it still suffers from modality misalignment, potentially overlooking visual input information. To address this, we introduce an image-response relevance score, $R_I(s)$, to calibrate the reward score $R_T(s)$. This score depicts the relevance between the generated sentence $s$ and input image $x_v$. We leverage CLIP-score [17] for this calculation, where the vision encoder in the CLIP model aligns with the vision encoder in the target LVLM. The image-response relevance score $R_I(s)$ is defined as:

$$R_I(s) = \max(100 * \cos(\mathcal{F}_I(x_v), \mathcal{F}_T(s)), 0), \tag{3}$$

where the $\mathcal{F}_I(x_v)$ and $\mathcal{F}_T(s)$ represent the visual CLIP embedding and textual CLIP embedding, respectively. Finally, the calibrated reward score $R(s)$ for the generated sentence $s$ is defined as:

$$R(s) = \lambda \cdot R_I(s) + (1 - \lambda) \cdot R_T(s), \tag{4}$$

where $\lambda$ is a hyperparameter used to balance the language instruction-following and image-response relevance scores. By combining both scores, we aim to redirect the attention of LVLM towards the input visual information, thus enhancing its modality alignment ability.

## 3.2 Iterative Fine-Tuning

After establishing the reward framework in CSR, we next discuss our iterative fine-tuning process. Within this framework, we iteratively perform two essential steps, namely candidate response generation and preference data curation and optimization. These steps are elaborated upon as follows:

### 3.2.1 Step-Level Candidate Response Generation

In candidate response generation, our objective is to generate responses to build preference data. To accomplish this, we employ a sentence-level beam search strategy. Initially, we concurrently sample multiple candidate sentences, utilizing the "end of sub-sentence" marker (e.g., "." in English) as the delimiter. Subsequently, for each sentence $s$, we compute its reward score $R(s)$ using Eqn. (4). From these scores, we then select the top-$k$ and bottom-$k$ sentences with the highest and lowest reward scores, respectively, to proceed to the subsequent round of sentence-level beam search. This iterative process continues until reaching the "end of response," conventionally represented as $\langle \text{eos} \rangle$. Once all sentences for a response $y = \{s_1, \cdots, s_{N_y}\}$ are generated, we calculate the cumulative reward score for the response as the sum of the reward scores for each sentence within it. This is

---

**Algorithm 1** Calibrated Self-Rewarding

---

**Require:** Dataset: $\mathcal{D} = \{x^{(i)}\}_{i=1}^{N}$; Reference model: $\pi_{\text{ref}}$; Number of iterations: $T$
 1: **for** $t = 1, \ldots, T$ **do**
 2:      **for** each $x \in \mathcal{D}$ **do**
 3:           **while** not reach the end of response **do**
 4:                Generate a bunch of candidate sentences from last-round sentences
 5:                **for** each candidate sentence $s$ **do**
 6:                     Compute the self-generated instruction-following score $R_T(s)$ by Eqn. (2)
 7:                     Calculate the image representation $\mathcal{F}_I(x_v)$ and sentence representation $\mathcal{F}_T(s)$
 8:                     Compute the image-response relevance score $R_I(s)$ by Eqn. (3)
 9:                     Compute the calibrated reward score $R(s)$ by Eqn. (4)
10:                Select top-k and bottom-k sentences with the highest and lowest reward scores
11:           Select the preferred response $y_{w,t}$ and dispreferred response $y_{l,t}$
12:      Update $\pi_\theta \leftarrow \arg\min_\theta \mathcal{L}_t(\pi_\theta; \pi_{\text{ref}})$, $\pi_{\text{ref}} \leftarrow \pi_\theta$.

---

defined as: $R(y) = \sum_{i=1}^{N_y} R(s_i)$, where $N_y$ is the number of sentences in response $y$. The detailed algorithm for candidate response generation is outlined in Algorithm 1.

### 3.2.2 Preference Curation and Optimization

After generating candidate responses with their reward scores, our next step is to curate preference dataset. Here, for each input prompt, we select the responses with the highest and lowest cumulative calibrated reward scores as the preferred and dispreferred responses, respectively, to construct the preference dataset for fine-tuning. For each iteration $t$, we denote the constructed preference data as: $\mathcal{D}_t = \{(x^{(i)}, y_{w,t}^{(i)}, y_{l,t}^{(i)})\}_{i=1}^{N}$. After obtaining the preference data, we fine-tune the target LVLM using DPO. At iteration $t$, we use the last iteration fine-tuned model $\pi_{\theta_{t-1}}$ as the reference model. Following Eqn (1), the loss at iteration $t$ of CSR is defined as:

$$\mathcal{L}_t = -\mathbb{E}_{(x, y_{w,t}, y_{l,t}) \sim \mathcal{D}} \left[ \log \sigma \left( \alpha \log \frac{\pi_\theta(y_{w,t}|x)}{\pi_{\theta_{t-1}}(y_{w,t}|x)} - \alpha \log \frac{\pi_\theta(y_{l,t}|x)}{\pi_{\theta_{t-1}}(y_{l,t}|x)} \right) \right]. \tag{5}$$

The training process of CSR is detailed in Algorithm 1.

## 4 Experiment

In this section, we empirically investigate CSR in addressing the modality misalignment problem of LVLMs, focusing on the following questions: (1) Can CSR help improve the performance of models on both comprehensive benchmarks and hallucination benchmarks? (2) Can CSR iteratively improve multimodal alignment progressively in LVLMs and lead to more factual LVLMs? (3) Is CSR compatible with different open-sourced LVLMs? (4) How does CSR change attention weights and preference pairs to align image and text modalities?

### 4.1 Experimental Setups

**Implementation Details.** We utilize LLaVA-1.5 7B and 13B [1] as the backbone models. During the preference learning process, we adapt LoRA fine-tuning [18]. The images and prompts used to construct the preference data are randomly sampled from the detailed description and complex reasoning subclasses of the LLaVA150k dataset, totaling approximately 13,000 samples [19]. It is worth noting that each iteration uses the same prompt and image as the previous round. Overall, the iterative training is conducted over three iterations, completed on one A100 80GB GPU. It takes roughly 3.5 and 5 hours for fine-tuning LLaVA-1.5 7B and LLaVA-1.5 13B, respectively. For more detailed information on training hyperparameters and training data, please refer to Appendix A.1.

**Evaluation Benchmarks.** We conducted evaluations on three types of benchmarks: comprehensive benchmarks, general VQA and hallucination benchmarks. Specifically, this includes: (1) Comprehensive benchmarks (MME [20], SEEDbench [21], LLaVA$^W$ [19], MMbench [22], MM-Vet [23]); (2) General VQA (ScienceQA (SQA) [24], VisWiz [25], GQA [26]); (3) Hallucination benchmark (POPE [27], CHAIR [28]). More detailed description are discussed in Appendix A.1.

Table 1: The performance of CSR on LLaVA-1.5 across all benchmarks is presented. Most baseline results, except those for self-rewarding, are sourced from Zhou et al. [10].

| Method | Comprehensive Benchmark | | | | | | General VQA | | | Hallucination Benchmark | | |
|---|---|---|---|---|---|---|---|---|---|---|---|---|
| | $MME^P$ | $MME^C$ | SEED | $LLaVA^W$ | MMB | MM-Vet | $SQA^I$ | VisWiz | GQA | POPE | $CHAIR_S$ | $CHAIR_I$ |
| LLaVA-1.5-7B | 1510.7 | 348.2 | 58.6 | 63.4 | 64.3 | 30.5 | 66.8 | 50.0 | 62.0 | 85.90 | 48.8 | 14.9 |
| + Vlfeedback | 1432.7 | 321.8 | 59.3 | 62.1 | 64.0 | 31.2 | 66.2 | 52.6 | **63.2** | 83.72 | 40.3 | 13.2 |
| + Human-Prefer | 1490.6 | 335.0 | 58.1 | 63.7 | 63.4 | 31.1 | 65.8 | 51.7 | 61.3 | 81.50 | 38.7 | 11.3 |
| + POVID | 1452.8 | 325.3 | 60.2 | 68.7 | 64.9 | 31.8 | 68.8 | 53.6 | 61.7 | 86.90 | 35.2 | 8.3 |
| + RLHF-V | 1489.2 | 349.4 | 60.1 | 65.4 | 63.6 | 30.9 | 67.1 | **54.2** | 62.1 | 86.20 | 29.7 | 7.5 |
| + Self-rewarding | 1505.6 | 362.5 | 60.0 | 61.2 | 64.5 | 31.4 | 69.6 | 53.9 | 61.7 | 86.88 | 24.0 | 6.7 |
| **+ CSR (Ours)** | **1524.2** | **367.9** | **60.3** | **71.1** | **65.4** | **33.9** | **70.7** | 54.1 | 62.3 | **87.01** | **21.0** | **6.0** |
| LLaVA-1.5-13B | **1531.3** | 295.4 | 61.6 | 70.7 | 67.7 | 35.4 | 71.6 | 53.6 | 63.3 | 85.90 | 48.3 | 14.1 |
| + Self-rewarding | 1529.0 | 300.1 | 62.8 | 65.6 | 64.5 | 35.3 | 74.3 | 56.1 | 63.2 | 86.58 | 37.0 | 8.8 |
| **+ CSR (Ours)** | 1530.6 | **303.9** | **62.9** | **74.7** | **68.8** | **37.8** | **75.1** | **56.8** | **63.7** | **87.30** | **28.0** | **7.3** |

**Baselines.** We will first compare CSR with the self-rewarding approach described by Yuan et al. [29]. Here, we directly apply self-rewarding to LVLM, using the prompts and experimental settings outlined in Yuan et al. [29] (see detailed settings in Appendix A.1 and Table 3). We also compared CSR with several data-driven preference learning methods, including Silkie (Vlfeedback) [9], LLaVA-RLHF (Human-preference) [8], POVID [10], and RLHF-V [30]. Furthermore, we compared the performance of the optimized LLaVA-1.5 via CSR with other state-of-the-art open-source LVLMs, including InstructBLIP [31], Qwen-VL-Chat [32], mPLUG-Owl2 [33], BLIP-2 [34], and IDEFICS [35], after the final rounds of training (CSR with iteration = 3). Additionally, to evaluate the effectiveness of CSR on other LVLMs, we applied CSR to a recent LVLM called Vila [36]. For more information on these baselines, please refer to Appendix A.1.

## 4.2 Results

**CSR Continuously Improves Model Performance over Iterations.** In Figure 3, we report the average performance of LLaVA-1.5 7B and 13B models concerning the number of training iterations on comprehensive benchmarks, general VQA tasks, and hallucination benchmarks. To facilitate score calculation, we first calculated an average score on a 100-point scale by adjusting the original values: $MME^P$ was divided by 16, and $MME^C$ was divided by 4, corresponding to the number of categories in MME. Additionally,

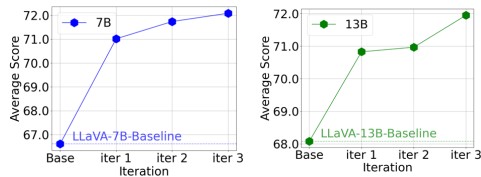

Figure 3: Average scores of CSR at different iterations over all benchmarks (see Table 6 and Table 7 in Appendix A.5 for full results).

since a lower CHAIR value indicates better performance, we standardized all metrics to follow a higher is better approach by transforming the $CHAIR_S$ and $CHAIR_I$ metrics into 100 - $CHAIR_S$ and 100 - $CHAIR_I$. We then calculated the average score by averaging these standardized values, which were used to compute the average percentage increase. In the experiment, the 7B model achieved an improvement of approximately 7.62% across all benchmarks through online iterative updates, while the 13B model saw an improvement of approximately 5.25%. According to the full results in Table 6 and Table 7 of Appendix A.5, the improvement is particularly significant on the $LLaVA^W$ and CHAIR benchmarks, with improvements of 8.9% and 49.50%, respectively. The results indicate that CSR is capable of incrementally improving model performance over iterations, demonstrating its effectiveness in self-improving the quality of generated preference data and leading to stronger modality alignment. The degree of improvement gradually becomes smaller, which is not surprising, indicating that the model is gradually converging.

**CSR Outperforms Competitive Preference Fine-Tuning Baselines.** Compared to preference data curation approaches (e.g., POVID, RHLF-V) that generate preference data from either additional models or human annotations, the superiority of CSR indicates that adapting a self-rewarding paradigm better captures the inherent preferences of the target LVLMs, achieving stronger modality alignment. Furthermore, CSR outperforms existing self-rewarding methods, with an average performance improvement of

Table 2: Ablation study of vision-text reward score.

| Method | 7B | 13B |
|---|---|---|
| Base | 66.61 | 68.08 |
| Only $R_T$ | 68.46 | 68.12 |
| Only $R_I$ | 67.49 | 69.23 |
| **CSR (Ours)** | **72.39** | **71.95** |

2.43%, demonstrating its effectiveness in calibrating the reward model by incorporating image-response relevance scores. This mitigates the potential issue of overlooking visual input information when estimating self-generated preferences.

In addition, we compare the performance of LLaVA-1.5 after three rounds of online CSR with other state-of-the-art open-sourced VLLMs and report the results in Table 5 of Appendix A.5. Although different open-sourced VLLMs utilize various image and text encoders, CSR still outperforms other open-sourced VLLMs in 9 out of 10 benchmarks, further corroborating the effectiveness of CSR in improving modality alignment.

### 4.3 Analysis

**Ablation Study.** To validate the effectiveness of using the image-response relevance score ($R_I$) to complement the self-generated instruction following score ($R_T$), we specifically compare CSR with three variants: (1) without applying CSR on LLaVA 1.5 (Base); (2) using CSR with only the self-generated instruction following score (Only $R_T$); and (3) using CSR with only the image-response relevance score (Only $R_I$). The results are reported in Table 2. We first observe that CSR improves performance by jointly considering both the self-generated instruction following and image-response relevance scores. This verifies its effectiveness in enhancing modality alignment by calibrating the language-driven self-rewarding paradigm with visual constraints. Additionally, we further conduct the analysis on the change of $\lambda$ in Eqn (4) and found that incorporating external visual scores to calibrate the models rewarding process effectively enhances performance (see detailed results in Appendix A.5.)

**Compatibility Analysis.** To validate CSR for its applicability to other LVLMs, we deployed CSR on Vila 7B and conducted three rounds of online iterations. We conducted experiments on all ten evaluation benchmarks and tasks, and the results are shown in Figure 4. Similar to the findings in Figure 3, Vila demonstrates a similar phenomenon during the online iterations of CSR, where it can self-correct preferences, leading to gradual improvements in all benchmarks. For Vila, the overall performance improved by 3.37% after three rounds of CSR iterations, with particularly notable increases of 8.48% on VisWiz and 14.0% on MM-Vet. The compatibility analysis further corroborates the generalizability and effectiveness of CSR in enhancing the performance of LVLMs.

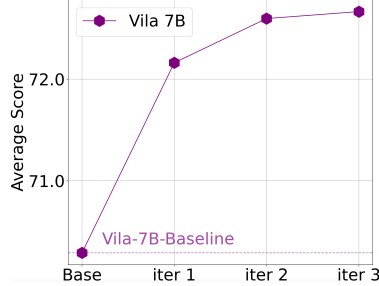

Figure 4: Average scores of CSR in Vila 7B at different iterations over all benchmarks (see Table 8 in Appendix A.5 for full results).

**How Does CSR Change the Image-Response Relevance Over Iterations?** To investigate how CSR gradually improve the performance over iterations, we analyzed the change of self-generated preference data with the LLaVA-1.5 7B model. In Figure 5, we illustrated the distribution of image-response relevance scores of three iterations over 500 examples from LLaVA-150k [19]. We first observe that both the chosen (preferred) and rejected (dispreferred) responses achieve higher image-response relevance scores after the model undergoes CSR online iterations. This indicates that, following CSR, the responses generated by LVLMs are more closely aligned with the image information. Secondly, it can be observed that after multiple rounds of online iterations with CSR, the average image-response relevance scores for the rejected and chosen responses become closer to each other. This makes the self-generated preference data during CSR iterations more challenging to distinguish, while further strengthening the learning process.

**How Does CSR Improve Modality Alignment?** To further understand how CSR affects modality alignment, in Figure 6, we present the changes in image and text attention maps for three models: the original LLaVA-1.5 7B model, the self-rewarding approach, and CSR. These attention maps illustrate the distribution of attention scores over image and text tokens. We observe that applying CSR strengthens the model's attention to certain visual tokens. Simultaneously, the change of attention values of the text tokens indicates that CSR is capable of alleviating

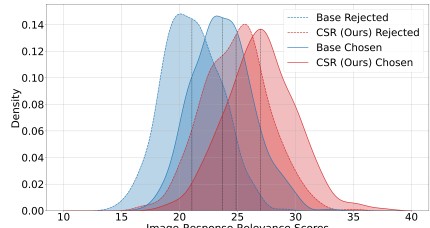

Figure 5: Image relevance scores before and after employing CSR.

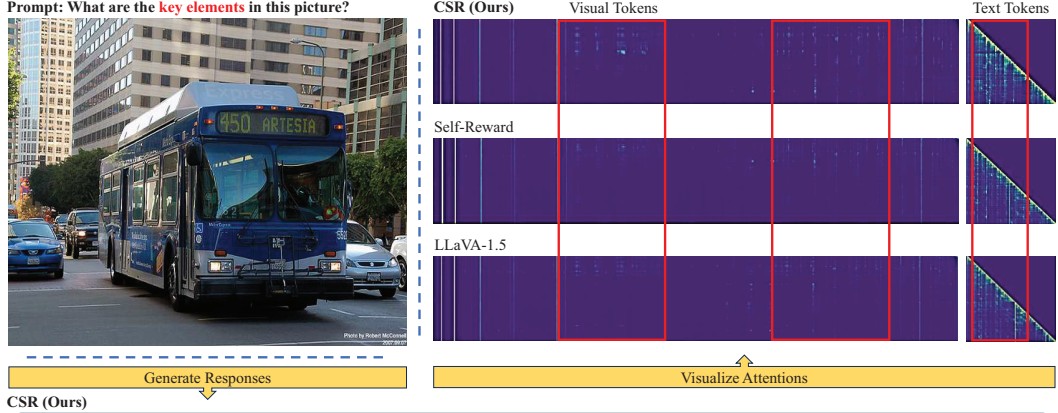

**Prompt: What are the key elements in this picture?**

**CSR (Ours)**

The image features a blue and white bus driving down a city street, with the bus's destination clearly displayed on the front: "Artesia." The bus is positioned in the middle of the street, and its large size and prominent destination sign make it easily noticeable to pedestrians and other vehicles. In addition to the bus, the image also includes a car driving behind the bus, indicating that both vehicles are sharing the roadway in the city.

**Self-Reward**

An articulated city bus, which is blue and white, is driving down a street in a large city. In front of the bus is a truck, and the bus is positioned behind it. Several people can be seen walking along or standing near both the bus and truck. They are likely pedestrians going about daily activities or waiting for public transportation. The scene captures the bustling atmosphere of a busy urban environment with various modes transportation in use. It highlights how public transportation, such as buses, plays a crucial role in the daily lives of city dwellers. Overall, it is a typical day in a large metropolitan area with people commuting and going about their daily routines.

**LLaVA-1.5**

The image features a blue and white bus driving down a city street, with a large building in the background. The bus is the main focus of the scene, occupying a significant portion of the image. A few cars can be seen on the street, with one car positioned behind and to the left of the bus, and another car located further down the street on the right side. There are several people in the scene, with one person standing near the left edge of the image, another person closer to the center, and a third person on the right side of the street. They appear to be pedestrians going about their daily activities. Additionally, a traffic light is visible in the background, indicating that the bus is driving in an urban environment with traffic regulations.

Figure 6: Comparison of attention maps. After optimizing the model with CSR, the attention scores allocated to visual tokens increase, indicating that CSR effectively redirects the model's attention toward the input visual information during the response generation process.

the issue of over-reliance on context mentioned in Huang et al. [37]. Additionally, compared with the self-rewarding approach, CSR shows a more effective distribution of attention between image and text tokens. These findings indicate that with CSR, LVLMs can better align different modalities through a calibrated self-rewarding strategy, focusing more on the visual modality rather than over-relying on contextual text.

### 4.4 Case Study

In this section, we use LLaVA-1.5 13B as an example to illustrate changes in the models own responses during CSR iterations and the preference data sampled in the CSR learning process, with hallucinations and errors highlighted in red. The results are shown in Figures 7 and 9, respectively. As shown in Figures 7, with each iteration of CSR, hallucinations in the models responses noticeably decrease. This indicates that CSR effectively refines the models preferences through iterative preference learning, making the models responses more accurate. In Figure 9 of the Appendix, we present a sampled preference data pair from the model during the CSR learning process. It can be seen that through CSR, the model not only gradually refines its own preferences but also obtains high-quality preference data pairs without human annotation.

## 5 Theoretical Explanation

In this section, we present a theoretical framework to explain the empirical phenomenon that incorporating an image-response relevance score can calibrate the self-rewarding procedure, ultimately improving generation accuracy.

As we consider an LVLM, to facilitate the analysis, we decompose the input prompt into $x = (x_v, x_t) \in \mathbb{R}^{d_v} \times \mathbb{R}^{d_t}$, representing the image and text prompts respectively. Although text data

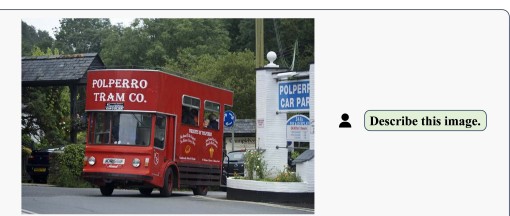

**Describe this image.**

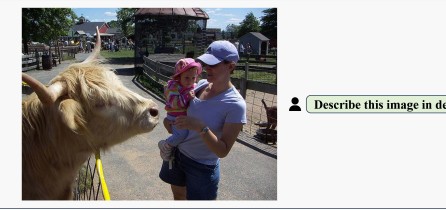

**Describe this image in detail.**

**CSR (Iteration = 3)**

The image features a red truck, referred to as a Polperro Tram Co. bus, traveling on the street with several passengers on board. The passengers, including adults and children, appear to be enjoying themselves as they ride along the tour. The bus is positioned in the center of the scene, taking up a substantial portion of the image.

Around the red bus, there are a few cars visible on the street, some in the foreground and others further behind the bus. The overall atmosphere of the image suggests a lively and engaging ride for the passengers, enjoying their time on the Polperro Tram Co. bus.

**CSR (Iteration = 2)**

The image showcases an old red bus with Polperro Tram Co. written on it, driving along a country route. The bus is loaded with passengers, including at least nine people visible through the windows, sitting and riding in a relaxed manner. As the bus travels down the street, it appears to be a charming and enjoyable trip for the passengers, with the lively red color of the bus adding a touch of personality to the rural setting.

**CSR (Iteration = 1)**

The image features a large, old-fashioned red bus, possibly a trolley, parked and waiting on a city street. The bus is packed with passengers, who can be seen through the windows, with at least nine people visible onboard. It appears that the bus is either picking up or dropping off passengers as it moves through the city.

The bus is parked in front of a building, surrounded by other vehicles nearby. There are three cars that can be spotted around the bus, suggesting the bus is in a bustling urban environment. The bus's vintage appearance adds charm and character to the scene, reflecting a bygone era of transportation.

**LLaVA-1.5-13B**

The image shows a red tram or bus parked on the side of a street with trees in the background, possibly in Europe. The bus is full of passengers waiting on the door area of the vehicle, while onlookers watch the scene from nearby. In total, eleven people can be seen either waiting on the bus or standing by the street.

In addition to the red tram, there are two cars visible in the scene. One car is behind and to the right of the tram, while the other car can be found further back on the right side of the street. There is also a handbag located close to one of the passengers, likely belonging to one of them.

**CSR (Iteration = 3)**

The image features a woman holding her child at a zoo, interacting with a large, long-haired animal like a yak or a bison. They are standing close to the animal, possibly feeding it or observing it up close. The woman and the child appear to be the center of attention in the scene, as they are focused on the impressive animal. The woman is wearing a blue shirt and shorts, which accent the family-oriented environment. The unique and close-up interaction between the mother, child, and the animal suggests a memorable and educational experience for the child.

**CSR (Iteration = 2)**

The image features a woman standing next to two children, who are holding onto her. The woman and two children appear to be at a zoo, interacting with an animal. In this particular scene, they are close to a Yak, which is a large, horned bovine with furry brown ears, creating an impressive sight for the children.

The woman, children, and Yak all share the same space, and it seems as if they are having a memorable and educational experience together at the zoo. The woman seems to be engaging with the kids in exploring the exhibit and sharing interesting information about the Yak, while the kids enjoy this unique encounter.

**CSR (Iteration = 1)**

The image depicts a family with a young child and a baby visiting a petting zoo, where they are tending to and feeding a large, furry brown animal, possibly a yak. The man holding the baby is smiling as he engages with the large furry animal. There are several people around them, who are also engaged in looking at and interacting with the animal, making the scene a joyful and lively atmosphere.

A bench can be seen further back in the scene, suggesting the zoo has a seating area for visitors to relax and enjoy the view. The presence of the large animal and the happy people surrounding it create a warm and friendly ambiance at the petting zoo.

**LLaVA-1.5-13B**

In the image, two little girls are being cuddled by their mother on a street. They seem to be enjoying their time together outside. In the background, a crowd of people can be seen, giving the impression of a busy place.

On the street, there are various objects such as benches and umbrellas scattered. Additionally, a clock is visible near the top part of the scene. The presence of the clock suggests that the location could be an area with shops or cafes. A handbag is noticeable belonging to one of the adults in the scene.

Figure 7: Two cases selected from the CSR-generated datasets.

typically comprises discrete tokens, we follow the CLIP theory literature [38–40] in modeling them as continuous-value random vectors in this section to elucidate the rationale behind our proposed method. More specifically, we assume the following data generative model for $x_v$ and $x_t$:

$$x_v = U_1 z_1 + \xi_1, \text{ and } x_t = U_2 z_2 + \xi_2,$$

where $U_1 \in \mathbb{O}^{d_v \times r}$ and $U_2 \in \mathbb{O}^{d_t \times r}$ are two orthonormal matrixces, representing decoders that transform the latent (low-dimensional) signals $z_1, z_2 \in \mathbb{R}^r$ to images and text respectively. We assume the covariance matrices of $z_1, z_2$ are identity matrices. $\xi_1 \in \mathbb{R}^{d_v}$ and $\xi_2 \in \mathbb{R}^{d_t}$ are noise vectors, and we assume they follow sub-gaussian distributions with well-conditioned covariance matrices and sub-gaussian norms upper bounded by a universal constant. We consider the infinite data setting. This is a widely used simplification to avoid the influence of sample randomness [41–43]. According to [38], with an abundance of image-text pairs, the learned visual CLIP embedding $\mathcal{F}_I(x_v)$ and textual CLIP embedding $\mathcal{F}_T(x_t)$ converge to $U_1^\top x_v$ and $U_2^\top x_t$ respectively. To simplify our analysis without loss of generality, we consider a single score for each response $y$ and define the image-response relevance score $R_I(y) = \langle U_1^\top x_v, U_2^\top y \rangle$.

We assume the ground truth $y_{truth} = V_1^* x_v + V_2^* x_t + \epsilon_y$ with weights $V_1^* \in \mathbb{R}^{d_v \times d_v}$ and $V_2^* \in \mathbb{R}^{d_v \times d_t}$. In CSR, we assume the conditional distribution at iteration $t$, $\pi_{\theta_t}(y \mid x)$ with $\theta_t = (V_1, V_2)$, follows a Gaussian distribution $\pi_{\theta_t}(y \mid x) \propto \exp(-\|y - (V_1 x_v + V_2 x_t)\|^2 / \sigma^2)$, where $V_1 \in \mathbb{R}^{d_v \times d_v}$ and $V_2 \in \mathbb{R}^{d_v \times d_t}$ are the weights matrices for the image and text inputs respectively, and $\sigma > 0$ is the standard deviation. As the likelihood is monotonically decreasing with respect to $\|y - (V_1 x_v + V_2 x_t)\|^2$, we consider the self-generated instruction-following score $R_T(y) = -\|y - (V_1 x_v + V_2 x_t)\|^2$. Then the calibrated reward score becomes $R(y) = \lambda \cdot R_I(y) + (1 - \lambda) \cdot R_T(y)$, for some $\lambda \in [0, 1]$. In theoretical analysis, we consider a simpler version of CSR, where we assume $y_w = \arg\max_y R(y)$ (whose distribution is denoted by $p_{\theta_t}^*(y \mid x)$), and $y_l$ is the text output generated by $\pi_{\theta_t}(y \mid x)$. As $R(y)$ depends on $\lambda$, we denote the solution $\theta_{t+1}$ by $\theta_{t+1}(\lambda)$. In the special case where $\lambda = 1$, this corresponds to the setting where we do not use the image-response relevance score at all.

To evaluate the quality of the text output $y$, we consider a regression problem where there is an outcome $z$ associated with the ground-truth text output $y_{truth}$: $z = \beta^{*\top} y_{truth}$ with $\beta^* \in \mathbb{R}^{d_t}$. We evaluate the quality of $y$ by considering the loss function $L(y) = \min_{\beta \in \mathbb{R}^{d_t}} \mathbb{E}[(z - \beta^\top y)^2]$. We then have the following theorem.

**Theorem 5.1.** *Suppose that $\pi_{\theta_t}^*(y \mid x)$ lies in the LLM space $\{\pi_\theta(y \mid x) : \theta \in \Theta\}$, $\|\beta^{*\top} V_1^{*\top} \beta^*\| \gg \|\beta^{*\top} V_2^{*\top} \beta^*\|$ and $\|\beta^{*\top} V_1^\top \beta^*\| \ll \|\beta^{*\top} V_2^\top \beta^*\|$, then there exists $\lambda < 1$, such that*

$$\mathbb{E}_{\pi_{\theta_{t+1}(\lambda)}(y|x)}[L(y)] < \mathbb{E}_{\pi_{\theta_{t+1}(1)}(y|x)}[L(y)].$$

Our theoretical analysis implies that as long as $\|\beta^{*\top} V_1^\top \beta^*\| \ll \|\beta^{*\top} V_2^\top \beta^*\|$, which happens when the model tends to prioritize textual information over visual input. By incorporating the image-response relevance score (corresponding to $\lambda < 1$), CSR is able to increase the attention on image signals in generating $y$. As a result, the solution produced by CSR will be better than the method without using the image-response relevance score (corresponding to $\lambda = 1$).

## 6 Related Work

**Large Visual-Language Model Hallucination.** Recently, the rapid development of visual-language alignment methods [19, 44–49] and LLMs [50–54] has significantly accelerated the progress of LVLMs, which extend LLMs with visual modalities and demonstrate impressive visual understanding by unifying the encoding of visual and text tokens [34, 55–57]. However, LVLMs still face the problem of hallucination [58, 59], where generated text descriptions contradict the visual modality information. Various approaches have been proposed to address hallucination in LVLMs, including enhancing dataset quality for fine-tuning [60, 8, 61, 9], manipulating the decoding process [37, 62–66], and leveraging external closed-source models to facilitate post-hoc mitigation of hallucination [58, 67–70]. Though these approaches alleviate hallucination to some extent, they do not focus directly on improving modality alignment.

**Preference and Modality Alignment.** In large models, alignment is necessary to ensure their behavior aligns with human preferences [71, 15, 72]. In LVLMs, alignment manifests as modality misalignment, where the generated textual responses are supposed to follow the input visual information. Recently, preference optimization has been used to address the modality misalignment problem. These optimizations involve preference data curated by human annotators [8, 60, 30] and additional models (e.g., GPT-4) [9, 10]. While these methods improve the ability of LVLMs to align modalities, their reliance on human annotation or additional models is resource-intensive and may introduce additional biases. Furthermore, these models cannot fully capture the inherent preferences of LVLMs, making the curated preference data less effective. Instead, CSR leverages a calibrated self-rewarding strategy, aiming to stimulate the LVLMs' self-correction and enhancement capabilities, thereby further improving modality alignment.

**Self-Improvement in Large Language Models.** Self-improvement emerges as a powerful paradigm for LLMs to enhance themselves without significant external intervention. For example, self-rewarding and online alignment propose a method that selects consistent answers generated by the model to fine-tune itself [73, 74], thereby improving its reasoning ability. Similarly, Chen et al. [12] utilizes self-play to enhance the model's performance by distinguishing its self-generated responses from those in human-annotated training data. Unlike prior methods that primarily target LLMs, CSR addresses the modality misalignment issue in LVLMs during the preference modeling process by introducing visual constraints, making it particularly well-suited for LVLMs.

## 7 Conclusion

In this paper, we investigate the challenge of enhancing modality alignment in LVLMs by introducing a calibrated self-rewarding approach, which integrates visual constraints into the preference modeling process of the self-rewarding paradigm. Empirically, CSR enhances the alignment between image and text modalities, significantly reducing hallucination and improving performance on various LVLM evaluation benchmarks. These empirical results are further supported by rigorous theoretical findings. Additionally, CSR is capable of continuously enhancing LVLM capabilities over iterations, leading to better utilization of visual information.

## Acknowledgement

We thank the Center for AI Safety for supporting our computing needs. This research was supported by Cisco Faculty Research Award.

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

# A  Additional Results

## A.1  Experimental Setup

### A.1.1  Hyperparameter Settings

**Sentence-Level Beam Search.** We configure our parameters as follows to ensure both diversity and quality in the sampled data. The num_beamsparameter, set to 5, determines the capacity of input at each search layer. Additionally, num_token_beams, also set to 5, ensures that each beam search returns 5 token-level search results. The eos_token_id is set to the token for a period, effectively controlling the sentence-by-sentence generation process. The max_length parameter, set to 1024, prevents truncation errors and infinite repetitions by controlling the maximum length, while max_new_tokens, set to 74, limits the maximum length of newly generated content to avoid exceeding the CLIP encoding limit.

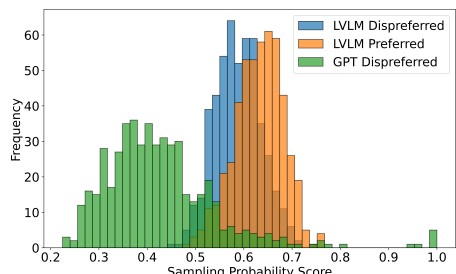

Figure 8: Distribution of preferred responses and dispreferred responses based on the sampling probability scores generated by LVLMs' language models.

To further enhance data diversity, we utilize group beam search by setting the num_beam_group parameter to 5. This approach, when matched with token-level search, significantly boosts the diversity of each data point. The diversity_penalty parameter, set to a value of 3.0, effectively controls the diversity and quality of the sampled data among different beam groups.

**Calibrated Rewarding.** We set the clip score weight to 0.9 and the language score weight to 0.1 when calculating the scores, giving greater emphasis to visual calibration.

## A.2  Evaluation Metrics and Benchmarks

- MME [20] is a comprehensive benchmark for assessing the capabilities of LVLMs in multimodal tasks. It systematically evaluates models across two primary dimensions: perception and cognition, through 14 meticulously designed subtasks that challenge the models' interpretative and analytical skills.

- SEED-Bench [21] is designed to evaluate the generative comprehension capabilities of LVLMs. It features an extensive dataset of 19K multiple-choice questions with precise human annotations, covering 12 distinct evaluation dimensions that assess both spatial and temporal understanding across image and video modalities.

- LLaVA$^W$ [19] is a comprehensive benchmark for evaluating visual reasoning models. It comprises 24 diverse images with a total of 60 questions, covering a range of scenarios from indoor and outdoor settings to abstract art.

- MMBench [22] introduces a dual-pronged approach: a meticulously curated dataset that significantly expands the scope and diversity of evaluation questions, and a pioneering CircularEval strategy that leverages ChatGPT to transform free-form predictions into structured choices.

- MM-Vet [23] is an evaluation benchmark tailored for assessing the multifaceted competencies of LVLMs. It systematically structures complex multimodal tasks into 16 distinct integrations derived from a combination of 6 core vision-language capabilities, providing a granular analysis of model performance across diverse question types and answer styles.

- ScienceQA [24] is a multimodal benchmark designed to evaluate and diagnose the multi-hop reasoning ability and interpretability of AI systems within the domain of science. It offers an expansive dataset of approximately 21k multiple-choice questions across a broad spectrum of scientific topics, complemented by detailed answer annotations, associated lectures, and explanations.

- VizWiz [25] is a dataset in the field of visual question answering (VQA), derived from a naturalistic setting with over 31,000 visual questions. It is distinguished by its goal-oriented approach, featuring images captured by blind individuals and accompanied by their spoken queries, along with crowdsourced answers.

- GQA [26] is a dataset engineered for advanced real-world visual reasoning, utilizing scene graph-based structures to generate 22 million diverse, semantically-programmed questions. It incorporates a novel evaluation metrics suite focused on consistency, grounding, and plausibility, establishing a rigorous standard for assessing in vision-language tasks.

- POPE [27] is an assessment methodology designed to scrutinize object hallucination in LVLMs. It reformulates the evaluation into a binary classification task, prompting LVLMs with straightforward Yes-or-No queries to identify hallucinated objects. POPE offers a stable and adaptable approach, utilizing various object sampling strategies to reveal model tendencies towards hallucination.

- CHAIR [28] is a widely-recognized tool for evaluating the incidence of object hallucination in image captioning tasks, which has two variants: $CHAIR_I$ and $CHAIR_S$, which assess object hallucination at the instance and sentence levels, respectively. Formulated as:

$$CHAIR_I = \frac{|\{\text{hallucinated objects}\}|}{|\{\text{all mentioned objects}\}|} \quad CHAIR_S = \frac{|\{\text{captions with hallucinated objects}\}|}{|\{\text{all captions}\}|}$$

Specifically, we randomly sampled 500 images from the COCO [75] validation set and evaluated object hallucination using the CHAIR metric.

## A.3 Overview of the Baselines

- LLaVA-1.5 [1] is an improvement based on the original LLaVA [19] model demonstrating exceptional performance and data efficiency through visual instruction tuning. It enhanced with a CLIP-ViT-L-336px visual backbone and MLP projection. By incorporating academic-task-oriented VQA data and simple response formatting prompts, LLaVA-1.5 achieves the state-of-the-art results at that time with a remarkably modest dataset of just 1.2 million public images.

- InstructBLIP [31] leverages instruction tuning on pretrained BLIP-2 models, integrating an instruction-aware Query Transformer to enhance feature extraction for diverse vision-language tasks. It achieved state-of-the-art zero-shot performance at the time across 13 datasets and excelled in fine-tuned downstream tasks, such as ScienceQA, showcasing its advantage over contemporaneous multimodal models.

- Qwen-VL-Chat [32] is built upon the Qwen-LM [4] with a specialized visual receptor and input-output interface. It is trained through a 3-stage process and enhanced with a multilingual multi-modal corpus, enabling advanced grounding and text-reading capabilities.

- mPLUG-Owl2 [33] employs a modular network design with a language decoder interface for unified modality management. It integrates shared modules for cross-modal collaboration and modality-adaptive components for feature retention, enhancing generalization in both text-only and multimodal tasks.

- BLIP-2 [34] is a vision-language pre-training framework that efficiently leverages off-the-shelf frozen image encoders and LLMs. Employing a two-stage pre-training strategy with a lightweight Querying Transformer, BLIP-2 bridges the modality gap between vision and language, enabling zero-shot image-to-text generation that adheres to natural language instructions while maintaining high compute-efficiency.

- IDEFICS [35] is an open-access visual language model that expands upon the Flamingo [44] architecture, offering both base and instructed variants with 9 billion and 80 billion parameter sizes. It is developed using solely publicly available data and models.

- POVID [10] is a novel training paradigm aligns the preferences of VLLMs through external preference data from GPT4 and the inherent hallucination patterns within the model triggered by noisy images.

- RLHF-V [30] collected fine-grained paragraph-level corrections from humans on hallucinations and performing dense direct preference optimization on the human feedback.

- Silkie [9] constructed a VLFeedback dataset using VLLMs annotation. Specifically, the responses were generated by 12 LVLMs models conditioned on multimodal instructions extracted from different datasets. The entire dataset was evaluated using GPT-4V to assess the generated outputs in terms of helpfulness, visual faithfulness, and ethical considerations. In this paper, the VLFeedback dataset was utilized to perform one round of DPO on LLaVA-1.5.

- LLaVA-RLHF [8] proposes a novel alignment algorithm called Factually Augmented RLHF, which enhances the reward model by incorporating additional factual information such as image captions and ground-truth multi-choice options. In this paper, the annotated preference data is used to conduct one round of preference learning on LLaVA1.5.

- Self-rewarding [29] introduces a method for self-feedback learning in LLMs and serves as a baseline for our approach, referred to as CSR. Specifically, for each input image and prompt, two outputs are sampled from LLaVA-1.5. The model is provided with the prompt mentioned in Table 3 and is tasked with determining which output is better. Finally, LLaVA-1.5 is fine-tuned using the collected preference data, with the entire setup and the images and prompts used for inference matching those of CSR.

## A.4 Do Different Sources of Preference Data Have Different Impacts?

The sources of preference data generally fall into two main categories: external preference data and self-generated data. External preference data typically represent preferences obtained from human annotations or GPT-4. Although external preference data generally have higher quality compared to self-generated data, are they really more effective? We conducted an analysis using 500 samples obtained from the original LLaVA-1.5 7B model. Following the same pipeline as CSR, we selected samples with the highest and lowest rewards as preferred (chosen) and dispreferred (rejected) responses. We further employed the GPT-4 API to transform preferred responses into dispreferred ones, with specific prompts referenced in Table 4.

In Figure 8, we present the distribution based on both the sampling probabilities score generated by the target LVLM, which describes the probability of the LVLM generating this response. Clearly, compared to the model's own generated dispreferred responses, the dispreferred responses modified by GPT-4V are not as easily confusable for the model. This result partially supports the idea that dispreferred responses generated by external models are more easily distinguishable by the target LVLM, making them less effective.

Table 3: Prompt for self-reward: utilizing the model itself as a judge to determine whether the corresponding response is a chosen response or a reject response.

---

Now you act as a judge, helping me determine which of the two texts I provide is closer to the given image and has fewer errors.
\*\*\*\*\*\*\*\*\*\*\*\*\*\*\*\*\*\*\*\*\*\*\*\*\*\*\*\*\*\*\*\*\*\*\*\*\*\*\*\*\*\*\*\*\*\*\*\*\*\*\*\*\*\*\*\*\*\*\*\*\*\*\*\*\*\*\*\*\*\*\*\*\*\*\*\*\*\*\*\*\*\*\*\*

**Response 1**:
{response 1}
**Response 2**:
{response 2}
\*\*\*\*\*\*\*\*\*\*\*\*\*\*\*\*\*\*\*\*\*\*\*\*\*\*\*\*\*\*\*\*\*\*\*\*\*\*\*\*\*\*\*\*\*\*\*\*\*\*\*\*\*\*\*\*\*\*\*\*\*\*\*\*\*\*\*\*\*\*\*\*\*\*\*\*\*\*\*\*\*\*\*\*

Please strictly follow the following format requirements when outputting, and don't have any other unnecessary words.
**Output Format**:
response 1 or response 2.

---

## A.5 Additional Experiments

In this subsection, we provide a comparison of CSR with other state-of-the-art models, a performance comparison of different CSR iterations, a comparison of hallucinations in different CSR iterations, validation experiments of CSR on other models, ablation study on $\lambda$ in Eqn (4), and the relationship between reward score and average performance score. Experiments strongly demonstrate the effectiveness of CSR.

For the ablation study on $\lambda$ in Eqn (4), our training settings are consistent with Table 1, with three rounds of iteration. The experimental results in Table 9 show that as the value of $\lambda$ increases, the models performance on various benchmarks improves. This suggests that calibrating the models

Table 4: Prompt for GPT-4 API: transform the provided response into negative ones based on the provided image.

| |
|---|
| Transform the provided response into negative ones based on the provided image. |
| ************************************************************************ |
| **Response**: |
| {chosen response from another LVLM or ground truth} |
| **Requirements**: |
| (1) Revise the response while maintaining its original format and order as much as possible. |
| (2) Based on the provided image, primarily add, replace, or modify entities in the input response to make them related to the image but incorrect. Adjust their attributes and logical relationships accordingly. |
| (3) The modifications in (2) must align with the image information, making the revised result difficult to discern. |
| ************************************************************************ |
| Please strictly follow the following format requirements when outputting, and don't have any other unnecessary words. |
| **Output Format**: |
| negative response |

Table 5: Comparison of LLaVA-1.5 with CSR and other open-sourced state-of-the-art LVLMs.

| Method | Comprehensive Benchmark | | | | | | General VQA | | |
| | $\text{MME}^P$ | $\text{MME}^C$ | SEED | $\text{LLaVA}^W$ | MMB | MM-Vet | $\text{SQA}^I$ | VisWiz | GQA |
|---|---|---|---|---|---|---|---|---|---|
| BLIP-2 | 1293.8 | 290.0 | 46.4 | 38.1 | - | 22.4 | 61.0 | 19.6 | 41.0 |
| InstructBLIP | 1212.8 | 291.8 | 53.4 | 60.9 | 36.0 | 26.2 | 60.5 | 34.5 | 49.2 |
| IDEFICS | 1177.3 | - | 45.0 | 45.0 | 48.2 | 30.0 | - | 35.5 | 38.4 |
| Qwen-VL-Chat | 1487.6 | 360.7 | 58.2 | 67.7 | 60.6 | **47.3** | 68.2 | 38.9 | 57.5 |
| mPLUG-Owl2 | 1450.2 | 313.2 | 57.8 | 59.9 | 64.5 | 36.2 | 68.7 | 54.5 | 56.1 |
| **CSR iter-3 7B** | 1524.2 | **367.9** | 60.3 | 71.1 | 65.4 | 33.9 | 70.7 | 54.1 | 62.3 |
| **CSR iter-3 13B** | **1530.6** | 303.9 | **62.9** | **74.7** | **68.8** | 37.8 | **75.1** | **56.8** | **63.7** |

Table 6: The performance of CSR online iteration with LLaVA-1.5 as the backbone on comprehensive benchmarks and general VQA.

| Method | Comprehensive Benchmark | | | | | | General VQA | | |
| | $\text{MME}^P$ | $\text{MME}^C$ | SEED | $\text{LLaVA}^W$ | MMB | MM-Vet | $\text{SQA}^I$ | VisWiz | GQA |
|---|---|---|---|---|---|---|---|---|---|
| LLaVA-1.5-7B | 1510.7 | 348.2 | 58.6 | 63.4 | 64.3 | 30.5 | 66.8 | 50.0 | 62.0 |
| **+ CSR iter-1** | 1500.6 | 367.5 | 60.4 | 69.7 | 64.7 | 32.2 | 70.3 | 54.0 | 62.1 |
| **+ CSR iter-2** | 1519.0 | **368.9** | 60.3 | 70.4 | 65.2 | 33.7 | 70.1 | 54.0 | 62.3 |
| **+ CSR iter-3** | 1524.2 | 367.9 | 60.3 | 71.1 | 65.4 | **33.9** | 70.7 | 54.1 | 62.3 |
| **+ CSR iter-4** | **1524.6** | 368.8 | 60.4 | 71.0 | 65.3 | 33.9 | 70.4 | 54.0 | 62.2 |
| **+ CSR iter-5** | 1520.1 | 367.2 | **60.5** | **71.3** | **65.4** | 33.8 | **70.8** | **54.2** | **62.4** |
| LLaVA-1.5-13B | 1531.3 | 295.4 | 61.6 | 70.7 | 67.7 | 35.4 | 71.6 | 53.6 | 63.3 |
| **+ CSR iter-1** | **1533.1** | 303.6 | 63.0 | 74.4 | 68.4 | 37.4 | 74.8 | 56.8 | 63.2 |
| **+ CSR iter-2** | 1530.4 | 301.1 | **63.0** | 74.3 | 68.5 | 37.2 | 75.0 | 56.0 | 63.2 |
| **+ CSR iter-3** | 1530.6 | **303.9** | 62.9 | **74.7** | **68.8** | **37.8** | 75.1 | **56.8** | 63.7 |
| **+ CSR iter-4** | 1530.4 | 301.4 | 63.0 | 74.2 | 68.3 | 37.3 | **75.2** | 56.6 | 63.4 |
| **+ CSR iter-5** | 1531.1 | 302.2 | 62.8 | 74.0 | 68.2 | 37.4 | 74.8 | 56.7 | **63.7** |

rewarding process using the visual score can enhance the preference learning process, thereby boosting performance.

Table 7: The performance of CSR online iteration with LLaVA-1.5 as the backbone on hallucination benchmarks.

| Method | POPE$_{acc}$ | POPE$_{f1}$ | CHAIR$_S$ | CHAIR$_I$ | Avg Length |
|---|---|---|---|---|---|
| | Hallucination Benchmark | | | | |
| LLaVA-1.5-7B | 85.90 | 84.29 | 48.8 | 14.9 | 89.03 |
| **+ CSR iter-1** | 86.94 | 85.80 | 26.6 | 7.2 | 80.59 |
| **+ CSR iter-2** | 86.82 | 85.62 | 23.0 | 6.1 | 82.62 |
| **+ CSR iter-3** | 87.01 | 85.93 | 21.0 | 6.0 | 83.29 |
| **+ CSR iter-4** | 87.05 | 85.95 | 19.0 | 5.9 | 81.34 |
| **+ CSR iter-5** | **87.16** | **85.98** | **18.3** | **5.4** | 82.07 |
| LLaVA-1.5-13B | 85.90 | 84.87 | 48.3 | 14.1 | 89.73 |
| **+ CSR iter-1** | 87.28 | 86.29 | 36.0 | 9.0 | 98.85 |
| **+ CSR iter-2** | **87.33** | 86.36 | 36.0 | 7.8 | 105.0 |
| **+ CSR iter-3** | 87.30 | 86.31 | 28.0 | 7.3 | 107.8 |
| **+ CSR iter-4** | 87.20 | **86.58** | **27.4** | 7.4 | 112.3 |
| **+ CSR iter-5** | 87.18 | 86.51 | 28.0 | **7.3** | 102.4 |

Table 8: The performance of CSR online iteration with Vila 7B as the backbone.

| Method | MME$^P$ | MME$^C$ | SEED | LLaVA$^W$ | MMB | MM-Vet | SQA$^I$ | VisWiz | GQA | POPE | CHAIR$_S$ | CHAIR$_I$ |
|---|---|---|---|---|---|---|---|---|---|---|---|---|
| | Comprehensive Benchmark | | | | | | General VQA | | | Hallucination Benchmark | | |
| Vila | 1533.0 | 316.4 | 61.1 | 69.7 | 68.9 | 34.9 | 68.2 | 57.8 | 62.3 | 85.50 | 31.0 | 8.8 |
| + CSR iter-1 | 1520.6 | 321.9 | 63.2 | 73.5 | **69.3** | 38.3 | 71.9 | 62.3 | 62.2 | 86.82 | 29.2 | **7.9** |
| + CSR iter-2 | 1536.0 | **322.6** | 63.4 | 74.2 | 69.1 | 39.7 | **72.3** | 62.6 | 62.1 | 87.30 | 28.2 | 8.0 |
| + CSR iter-3 | **1542.2** | 321.5 | **63.4** | **74.3** | **69.3** | **39.8** | 72.2 | **62.7** | **62.4** | **87.31** | **28.0** | 8.2 |

Table 9: Performance comparison of CSR on LLaVA-1.5 7B with different $\lambda$ values on various benchmarks.

| Method | MME$^P$ | MME$^C$ | SEED | LLaVA$^W$ | MMB | MM-Vet | SQA$^I$ | VisWiz | GQA | POPE | CHAIR$_S$ | CHAIR$_I$ |
|---|---|---|---|---|---|---|---|---|---|---|---|---|
| ($\lambda = 0.1$) | 1508.6 | **369.3** | 60.0 | 66.7 | 64.9 | 31.6 | 70.0 | 54.0 | 62.0 | 86.90 | 40.8 | 10.2 |
| ($\lambda = 0.5$) | 1515.4 | 364.5 | 60.1 | 68.2 | 64.9 | 32.4 | 69.7 | 54.0 | 62.1 | 86.90 | 28.2 | 6.7 |
| ($\lambda = 0.9$) | **1524.2** | 367.9 | **60.3** | **71.1** | **65.4** | **33.9** | **70.7** | **54.1** | **62.3** | **87.01** | **21.0** | **6.0** |

Table 10: Reward score and average performance score across multiple iterations of CSR on LLaVA-1.5 7B.

| Iteration | Iter-1 | Iter-2 | Iter-3 | Iter-4 | Iter-5 |
|---|---|---|---|---|---|
| Chosen reward | 0.4885 | 0.5040 | 0.5052 | 0.5055 | 0.5066 |
| Rejected reward | 0.4551 | 0.4788 | 0.4789 | 0.4794 | 0.4799 |
| Avg performance score | 66.61 | 71.02 | 71.74 | 72.09 | 72.24 |

# B  Proofs

*Proof of Theorem 5.1.* Let us first denote the distribution of $y_w$ by $\pi^*_{\theta_t}(y \mid x)$. As we take $y_w = \arg\max_y R(y)$, this distribution is a point mass. As a result, the global minimizer to (5) will then converge to $\pi^*_{\theta_t}(y \mid x)$.

In the following, we analyze how $\pi^*_{\theta_t}(y \mid x)$ is shaped.

By the CSR procedure, we have

$$y_w = \arg\max_y (1 - \lambda))\langle U_1^\top x_v, U_1^\top y\rangle - \lambda\|y - V_1 x_v + V_2 x_t\|^2 = \frac{1-\lambda}{\lambda}U_1 U_1^\top x_v + V_1 x_v + V_2 x_t.$$

We can see that CSR up-weights the signal of the image input.

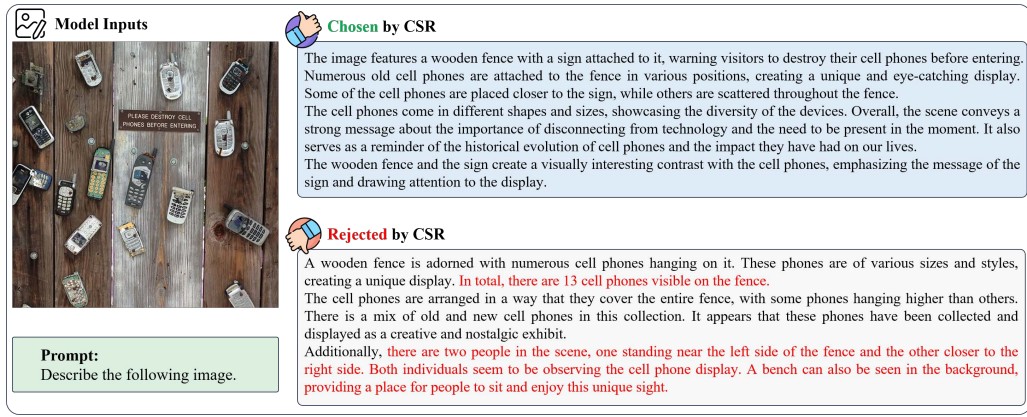

Figure 9: A case including both self-generated preferred and dispreferred responses.

Then

$$L(y) = \min_{\beta \in \mathbb{R}^{d_t}} \mathbb{E}[(z - \beta^\top y)^2] = \min_{\beta \in \mathbb{R}^{d_t}} \mathbb{E}[(\beta^{*\top} y_{truth} - \beta^\top y)^2]$$
$$= \min_{\beta \in \mathbb{R}^{d_t}} \mathbb{E}[(\beta^{*\top}(V_1^* x_v + V_2^* x_t)) - \beta^\top y)^2] + Var(\epsilon_y)\|\beta^*\|^2$$

We have

$$\mathbb{E}[(\beta^{*\top}(V_1^* x_v + V_2^* x_t)) - \beta^\top y)^2] = \mathbb{E}[(\beta^{*\top}(V_1^* x_v + V_2^* x_t))$$
$$- \beta^\top \left( (\frac{1-\lambda}{\lambda} U_1 U_1^\top + V_1) x_v + V_2 x_t \right))^2]$$

As we assume $\frac{\|V_1\|}{\|\beta^{*\top} V_1^*\|} \ll \frac{\|V_2\|}{\|\beta^{*\top} V_2^*\|}$ and due to the smoothness over parameters. Without loss of generality, we prove the claim for the case where $\|V_1\| = 0$, that is $V_1 = 0$.

In this case, we want to show that there exists $\lambda \in (0,1)$, such that

$$\min_{\beta \in \mathbb{R}^{d_t}} \mathbb{E}[(\beta^{*\top}(V_1^* x_v + V_2^* x_t)) - \beta^\top \left( (\frac{1-\lambda}{\lambda} U_1 U_1^\top) x_v + V_2 x_t \right))^2]$$
$$< \min_{\beta \in \mathbb{R}^{d_t}} \mathbb{E}[(\beta^{*\top}(V_1^* x_v + V_2^* x_t)) - \beta^\top (V_2 x_t))^2]$$

Due to the independence between $x_v$ and $x_t$, the right-hand sides is lower bounded by $\beta^{*\top} V_1^* Cov(x_t) V_1^{*\top} \beta^*$.

The left-hand side, on the other hand, can be upper bounded by the value when we take $\beta_0$ such that $\frac{1-\lambda}{\lambda} U_1 U_1^\top \beta_0 = U_1 U_1^\top V_1^{*\top} \beta^*$, which equals to $\beta^{*\top} V_1^* (I - U_1 U_1^\top) Cov(x_t)(I - U_1 U_1^\top) V_1^{*\top} \beta^*$.

As we assume $\|\beta^{*\top} V_1^{*\top} \beta^*\| \gg \|\beta^{*\top} V_2^{*\top} \beta^*\|$, this is a dominating term when the left-hand side is evaluated at $\beta_0$.

In addition, we assume $Cov(\xi_1)$ is well-conditioned, implying $Cov(x_t)$ is well-conditioned, and therefore

$$\beta^{*\top} V_1^* (I - U_1 U_1^\top) Cov(x_t)(I - U_1 U_1^\top) V_1^{*\top} \beta^* < \beta^{*\top} V_1^* Cov(x_t) V_1^{*\top} \beta^*.$$

We complete the proof.

$\square$

## C  Limitations

Due to limitations in computing resources, we conducted only three iterations of CSR. Additionally, our experiments were confined to 7B and 13B models. This restriction prevents us from determiing

if our method adheres to a scaling law. We hope to continue iterative training in the future and to train larger models, given access to more computing resources, to explore the upper limits of our method.

## D  Broader Impacts

Our approach requires no additional human annotations and significantly enhances model performance using the model itself. Technically, our method may inspire more researchers to explore how multimodal models can learn from themselves. From a societal impact perspective, our method significantly reduces hallucinations in LVLMs, a major factor affecting the application of AI in real-world scenarios. Our approach promotes more responsible use of LVLMs. However, it is important to note that while our method significantly reduces hallucinations, they still occur. Therefore, it is crucial to employ various measures to ensure safety and stability when applying this approach in real-world scenarios.

