# OpenReview forum: "Calibrated Self-Rewarding Vision Language Models"
_NeurIPS.cc/2024/Conference — NeurIPS 2024 poster_

### Official Review · Reviewer_4vrA · 2024-07-10

**Soundness:** 4
**Presentation:** 4
**Contribution:** 4
**Rating:** 8
**Confidence:** 3

**Summary:**

This paper addresses an important and tough issue in LVLM – hallucination, which is usually caused by the misalignment of the image and text modalities. A new method CSR is proposed, by extending the language self-awarding approach to multimodality, considering both instruction-following (text) and  image-text alignment (multimodality) awarding.

The experiments are very comprehensive and solid, demonstrating the effectiveness of CSR in a wide range of tasks, especially for hallucination specific tasks.

**Strengths:**

The issue this paper addresses is important and still an open question for LVLM training. The method proposed by this paper is simple yet effective, thus may be very insightful for the community.

The evaluation is comprehensive and solid, including 3 types of benchmarks: comprehensive benchmarks, general VQA and hallucination benchmarks, with 10 tasks in total. CSR is also compared with self-rewarding and several data-driven preference learning methods to demonstrate its effectiveness. Also different model sizes and model architectures are verified. The addtional Theoretical Explanation and attention map sections make the paper even more convincing.

The proposed method is easy to implement and extend to other modalities, so may have larger impact for other directions in the community.

This paper is very well written and readable.

**Weaknesses:**

The ablation of the weight of instruction-following score and image-text alignment score is missing, which is important to help understand how important of each score for CSR.

It might worth inverstigating the impact of different image-text alignment scoring approaches. At least comparing different image-text models.

**Questions:**

It would be more insightful if we can distinguish 1) comprehensive, 2) general VQA and 3) hallucination scores (e.g. avg score per type of benchmarks) in the main ablation, i.e., Table 2. I would expect “Only RI” to increase 3) but hurt 1) and 2); however, this is not verifiable given the current average scores.

The terminologies “sentence group” and “option” in Figure 2 are confusing (especially before readding 3.2). More legend or explanation (or re-structuring the figure) might be needed for better readability.

What is the consideration to use the same examples in different iterations? Using random examples may lead to even better results (to converge later)?

---

> ### Author Rebuttal · Authors · 2024-08-07
>
> Thank you for your valuable feedback. We've addressed your questions below and would appreciate it if you could let us know whether our responses meet your expectations.
>
> > **Q1**: The ablation of the weight of instruction-following score and image-text alignment score is missing.
>
> **A1**: We evaluate the effectiveness of varying $\lambda$ on the performance. The results are shown in Table R1 and indicate that placing greater emphasis on visual calibration significantly improves performance, further strengthening our contribution to reward calibration.
>
> **Table R1**: Performance vs $\lambda$
> | Category            | VQA  | Hallucination | Comp_bench |
> |-------------------|-------|------|------|
> | CSR-7B  ($\lambda$=0.9)     | 62.4|86.7| 69.7 |
> | CSR-7B  ($\lambda$=0.5)     | 61.9|84.0| 68.6 |
> | CSR-7B  ($\lambda$=0.1)     | 62.0|78.6| 68.3 |
>
> ---
>
> > **Q2**: Inverstigating the impact of different image-text alignment scoring approaches.
>
> **A2**: To further demonstrate the compatibility of CSR, we conducted an experiment using GPT-4o to calculate the image-response relevance score, which offers stronger image perception capabilities. We used the first iteration of CSR training on LLaVA-1.5-7B as an example, and the results are shown in Table R2. The results indicate that using a model with stronger image perception capabilities to calibrate the initial reward can provide improvements. We plan to include the performance of using more vision-centric models to calibrate the initial reward in the next version.
>
> **Table R2**: Comparison between different models that are used to calculate image-response relevance score
> | Method       | MME$^P$   | MME$^C$  | SEED  | LLaVA$^W$ | MMB   | MM-Vet | SQA$^I$  | VisWiz | GQA   | POPE  | CHAIR$_S$ | CHAIR$_I$ |
> |---------------|--------|-------|-------|--------|-------|--------|-------|--------|-------|-------|--------|--------|
> | CSR iter-1    | 1500.6 | 367.5 | 60.4  | 69.7   | 64.7  | 32.2   | 70.3  | 54.0   | 62.1  | 86.94 | 26.6   | 7.2    |
> | CSR iter-1 (GPT-4o)    | 1509.4 | 366.2 | 60.4  | 70.2   | 65.1  | 31.8   | 70.4  | 54.0   | 62.2  | 87.11 | 24.2   | 6.6    |
>
> ---
>
> > **Q3**:  Distinguish 1) comprehensive, 2) general VQA and 3) hallucination scores in the main ablation.
>
> **A3**: We have split the ablation results and reported them in Table R3. The results indicate that relying solely on the image-response relevance score increases the hallucination score and slightly negatively impacts, or at best, only marginally improves, the VQA and comprehensive benchmarks (likely affecting only the VQA performance in these benchmarks).
>
> **Table R3**: Fine-grained results of ablation study
> | Category            | VQA  | Hallucination | Comp_bench |
> |-------------------|-------|------|------|
> | LLaVA-1.5-7B     | 59.6| 74.1|66.4 |
> | Only $R_T$     | 60.0|75.0| 69.4 |
> | Only $R_I$     | 59.1 | 76.7 | 67.1 |
> | CSR (Ours) 7B     |62.4| 86.7 | 69.7 |
> |-------------------|-------|------|------|
> | LLaVA-1.5-13B  | 62.8| 74.5| 67.5 |
> | Only $R_T$   |63.4| 74.7| 67.2 |
> | Only $R_I$      |62.7 | 77.0 | 68.6 |
> | CSR (Ours) 13B    |65.2| 84.0| 69.3 |
>
> ---
>
> > **Q4**: The terminologies “sentence group” and “option” in Figure 2 are confusing.
>
> **A4**: We have polished Figure 2 and put it in the supplementary PDF (see Figure R2). Here is an explanation of this figure: Since CSR uses step-level reward modeling, we initially generate a few sentences (five for illustration). Each of these sentences is then used to sample five possible next sentences, resulting in a total of 25 sentences, which is named **“Option”**. To improve efficiency, only the top 3 sentences (indicated by green dots in the figure) and the bottom 2 sentences (indicated by red dots) are retained, which is named as **“Sentence Group”**. The retained five sentences are then used to generate the third step sentence, and this process is repeated until the response is complete.
>
> ---
>
> > **Q5**: What is the consideration to use the same examples in different iterations?
>
> **A5**: In CSR, we use the same samples for iterative learning for two main reasons:
>
> - A single round of preference learning may not fully resolve all issues. Multiple iterations allow for the incremental correction of the model's errors on these samples, leading to further improvements in model performance.
>
> - Using the same samples helps us investigate whether the model's responses to these samples improve across different iterations. This consistency is useful for analytical experiments (e.g. Figure 5).
>
> Additionally, we conducted an experiment during the second iteration of CSR with LLaVA-1.5-7B, where we selected 13,000 images that were entirely distinct from those used in the first iteration. The results of this experiment are reported in Table R4. It can be observed that using different samples across iterations performs similarly to using the same samples. We plan to conduct further investigations on this topic in the next version.
>
> **Table R4**: Results w/ random image batch over different iterations
> | CLIP model        | MME$^P$   | MME$^C$  | SEED  | LLaVA$^W$ | MMB   | MM-Vet | SQA$^I$  | VisWiz | GQA   | POPE  | CHAIR$_S$ | CHAIR$_I$ |
> |-------------------|-------|------|------|--------|------|--------|------|--------|------|------|--------|--------|
> | CSR iter-1  | 1500.6| 367.5| **60.4** | 69.7   | 64.5 | 32.2   | 70.3 | 54.0   | 62.1 | **86.94** | 26.6   | 7.2    |
> | CSR iter-2     | **1519.0** | 368.9 | 60.3 | 70.4   | **65.2** | **33.7**   | 70.1 | 54.0   | **62.3** | 86.82 | 23.0   | 6.1    |
> | CSR iter-2 (**other data**)     | 1513.4 | **369.1** | 60.2 | **70.7**   | 64.9 | 33.6   | **70.5** | **54.1**   | 62.2 | 86.90 | **21.3**   | **5.8**    |

---

> > ### Comment · Reviewer_4vrA · 2024-08-09
> >
> > Thanks for the updates and the detailed explanation, which have addressed all my questions. I am happy to maintain the previous rating.

---

> > > ### Author Response · Authors · 2024-08-09
> > > **Response to Reviewer 4vrA**
> > >
> > > Dear Reviewer 4vrA,
> > >
> > > Thank you for your response. We are pleased to hear that our answers have addressed all of your questions.

---

### Official Review · Reviewer_bHbK · 2024-07-13

**Soundness:** 3
**Presentation:** 3
**Contribution:** 2
**Rating:** 5
**Confidence:** 3

**Summary:**

The paper generally follows self-rewarding language models and applies the idea to vision-language models. The method first ask a VLM to self-generate candidates, based on which they score the candidates with the VLM itself and CLIPScore, and then perform DPO on the generated candidates. Experiments on LLaVA demonstrate improvements over baselines such as self-rewarding.

**Strengths:**

1. The paper investigates the application self-rewarding proposed in NLP to the vision-language domain and proposes VL-specific changes.
2. The baselines are fairly strong and they perform evaluations on multiple standard benchmarks, making their results trustworthy.
3. The paper is well-written.

**Weaknesses:**

1. The methodological contribution is not sufficient as it basically follows self-rewarding language models [11] and applies it in another domain. The proposed techniques such as using CLIPScore to calibrate the self-generated scores are incremental changes to me.
2. The empirical results do not seem significant in many of the datasets. In Table 1, on popular benchmarks such as GQA and SEED, the performance improvements are marginal (one exception may be LLaVA-W, but this dataset is small and relies on GPT evaluations). Results on VQAv2 and MMStar should also be added.

**Questions:**

Please refer to the weakness section.

**Limitations:**

the authors adequately addressed the limitations

---

> ### Author Rebuttal · Authors · 2024-08-07
>
> Thank you for your valuable feedback. We have answered your questions below, and we would appreciate it if you could let us know whether our responses address your concerns.
>
> > **Q1**: The methodological contribution is not sufficient as it basically follows self-rewarding language models [11] and applies it in another domain. The proposed techniques such as using CLIPScore to calibrate the self-generated scores are incremental changes to me.
>
> **A1**: We would like to highlight our contribution in comparison to LLM self-rewarding. Compared with self-rewarding in LLM, the fundamental difference in LVLM is that using the target LVLM as both the generator and judge (as is done in LLM self-rewarding) can amplify the misalignment issues between text and image modalities. This happens because the judge itself also tends to be biased toward the contextual information, often ignoring the input image. By incorporating visual constraints, we can effectively mitigate this issue, as it encourages better alignment between text and image modalities within the judge. We believe this is the fundamental difference between our approach and LLM self-rewarding. We believe these contributions are significant.
>
> ---
>
> > **Q2**: The empirical results do not seem significant in many of the datasets. In Table 1, on popular benchmarks such as GQA and SEED, the performance improvements are marginal (one exception may be LLaVA-W, but this dataset is small and relies on GPT evaluations). Results on VQAv2 and MMStar should also be added.
>
> **A2**: We would like to emphasize that the primary goal of CSR is to enhance the alignment between image and text modalities and minimize hallucination. CSR has demonstrated substantial improvements over other methods across the majority of the datasets evaluated. These improvements are significant; we achieved average improvements of 8.23% and 2.59% compared to the original 7B model and the strongest baseline (i.e., self-rewarding), respectively.
>
> To further evaluate the effectiveness of CSR, we evaluated LLaVA-1.5 7B and 13B of CSR for VQAv2 and MMStar, following the setup in Table 6. The results are shown in Table R1, including the original LLaVA, the strongest baseline (self-rewarding), and CSR. The performance gain of CSR on these additional benchmarks further demonstrates the effectiveness of CSR.
>
> **Table R1**: Performance on VQAv2 and MMStar
> | Model              | VQAv2   | MMStar (Avg performance) |
> |---------------------|--------------|--------------|
> | LLaVA-1.5-7B           | 78.5         | 30.3 |
> | +Self-rewarding  | 78.5          | 31.5 |
> | **+CSR**  | **78.7**          | **34.0**  |
> |---------------------|--------------|--------------|
> | LLaVA-1.5-13B           | 80.0         | 32.8 |
> | +Self-rewarding  | 80.2          | 34.7  |
> | **+CSR**   | **80.5**          | **36.6**  |
>
> ---
>
> **References**
>
> [1] Zhou Y, Cui C, Rafailov R, et al. Aligning modalities in vision large language models via preference fine-tuning[J]. arXiv preprint arXiv:2402.11411, 2024.
>
> [2] Yu T, Yao Y, Zhang H, et al. Rlhf-v: Towards trustworthy mllms via behavior alignment from fine-grained correctional human feedback[C]//Proceedings of the IEEE/CVF Conference on Computer Vision and Pattern Recognition. 2024: 13807-13816.
>
> [3] Sun Z, Shen S, Cao S, et al. Aligning large multimodal models with factually augmented rlhf[J]. arXiv preprint arXiv:2309.14525, 2023.
>
> [4] Li L, Xie Z, Li M, et al. Silkie: Preference distillation for large visual language models[J]. arXiv preprint arXiv:2312.10665, 2023.

---

> > ### Comment · Area_Chair_XWxf · 2024-08-11
> > **Please discuss with authors**
> >
> > Dear Reviewer bHbK:
> >
> > Please respond to author rebuttal and discuss with authors.
> >
> > Thanks,
> >
> > Your AC

---

> > ### Comment · Reviewer_bHbK · 2024-08-11
> > **Thank you for the response!**
> >
> > Thank you for your response!
> >
> > I've read the rebuttal and the other reviews.
> >
> > My concern about the effectiveness of the method is mostly addressed (though the improvements on VQAv2 seem rather marginal.)
> >
> > Regarding the technical contribution, I agree that using an LVLM as a judge may be biased toward the contextual information. Although interpolating the CLIPScore may not be a perfect solution, the empirical improvements suggest it is a simple and effective approach, even though less principled to me.
> >
> > Considering these factors and the other positive reviews, I will raise my score to borderline accept.

---

> ### Author Response · Authors · 2024-08-12
> **Response to Reviewer bHbK**
>
> Dear Reviewer bHbK,
>
> Thank you for your response and for increasing your score. We're delighted that our answers addressed your questions. We'll continue exploring ways to further calibrate reward scores in the future.

---

### Official Review · Reviewer_myph · 2024-07-13

**Soundness:** 3
**Presentation:** 3
**Contribution:** 2
**Rating:** 7
**Confidence:** 4

**Summary:**

The paper addresses the challenge of hallucination in Large Vision-Language Models (LVLMs), where generated text responses appear plausible but contradict the input image. This misalignment occurs because the models prioritize textual information over visual input, even with high-quality representations. Existing methods to address this issue involve resource-intensive preference optimization through additional models or human annotations, which may not align well with the LVLM’s preferences.

To overcome these challenges, the authors propose the Calibrated Self-Rewarding (CSR) approach. CSR allows the model to self-improve by generating candidate responses, evaluating rewards for each, and curating preference data for fine-tuning. The process emphasizes visual input through a step-wise strategy and incorporates visual constraints into the self-rewarding mechanism.

**Strengths:**

* The proposed method reduces the resources required to align a VLM model for less hallucination compared to previous methods.

* The paper is clearly written.

* The authors show theoretical explanation to validate the proposed method.

**Weaknesses:**

* In the limitation section, the authors mention that they could only run three iterations due to computation issues. However, in Section 4.1, the authors say they used one A100 80GB GPU, which takes roughly 3.5 and 5 hours to fine-tune LLaVA-1.5 7B and LLaVA-1.5 13B, respectively. It seems like such an experiment could be done with more iterations to see how the score trend continues across more iterations. This could supplement the claim of increased performance concerning iterations made in Section 4.2.

* The authors use the CLIP similarity score to align the image and text response. However, there is a concern that CLIP may primarily focus on prominent objects in the foreground, potentially overlooking smaller details in the background. If this is the case, using CLIP for reward calculation might inadvertently cause the Vision-Language Model (VLM) to miss fine details in the image, even though it would reduce hallucinations overall.

* Although Table 2 shows an analysis of the effect of each reward term ($R_I$ and $R_T$), it would be beneficial to see how varying $\lambda$ terms affect the score.

* The benefit of the proposed method, apart from the resulting benchmark score, is that it takes much less resources to align the model. Can a rough estimate of how different the computing resource would be, including the time to gather the preference data for previous methods?

* (minor) To this reviewer, the overall framework figure (Figure 2) seems hard to understand even after understanding the proposed method through the texts in the paper.

**Questions:**

See weaknesses above

**Limitations:**

The authors addressed the limitations of the work.

---

> ### Author Rebuttal · Authors · 2024-08-07
>
> Thank you for your constructive comments. Below are our responses to your questions. Please let us know if they address your concerns.
>
> > **Q1**: In the limitation section, the authors mention that they could only run three iterations due to computation issues. However, in Section 4.1, … supplement the claim of increased performance concerning iterations made in Section 4.2.
>
> **A1**: Using LLaVA-7B as an example, we performed two additional rounds of iterative training on top of the three rounds of CSR training detailed in Table 7. The results are presented in Table R1. According to the results, we observe that more iterations can still improve model performance, especially in mitigating hallucination. However, the overall improvement tends to slow down with additional iterations (see Figure R5 in Supplementary PDF), indicating performance convergence.
>
> **Table R1**: Results of more rounds of evaluation
> | Method        | MME$^P$   | MME$^C$  | SEED  | LLaVA$^W$ | MMB   | MM-Vet | SQA$^I$  | VisWiz | GQA   | POPE  | CHAIR$_S$ | CHAIR$_I$ |
> |---------------|--------|-------|-------|--------|-------|--------|-------|--------|-------|-------|--------|--------|
> | CSR iter-3  | 1524.2 | 367.9 | 60.3  | 71.1   | 65.4  | 33.9   | 70.7  | 54.1   | 62.3  | 87.01 | 21.0   | 6.0    |
> | CSR iter-4  | 1524.6 | 368.8 | 60.4  | 71.0   | 65.3  | 33.9   | 70.4  | 54.0   | 62.2  | 87.05 | 19.0   | 5.9    |
> | CSR iter-5  | 1520.1 | 367.2 | 60.5  | 71.3   | 65.4  | 33.8   | 70.8  | 54.2   | 62.4  | 87.16 | 18.3   | 5.4    |
>
> ---
>
> > **Q2**: Concern about the CLIP model in capturing details.
>
> **A2**: First, we would like to clarify that our proposed CSR framework is general enough to incorporate different vision-centric models used to calculate image-response relevance scores, even though CLIP is used in this paper. To further demonstrate the compatibility of CSR, we conducted an experiment using GPT-4o to calculate the image-response relevance score, which offers stronger image perception capabilities. We used the first iteration of CSR training on LLaVA-1.5-7B as an example, and the results are shown in Table R2. The results indicate that using a model with stronger image perception capabilities to calibrate the initial reward can provide improvements compared to using CLIP, although CLIP is sufficiently strong and lightweight. We plan to include the performance of using more vision-centric models to calibrate the initial reward in a future version.
>
> **Table R2**: Comparison of different models used to calculate image-response relevance scores
>
> | Method       | MME$^P$   | MME$^C$  | SEED  | LLaVA$^W$ | MMB   | MM-Vet | SQA$^I$  | VisWiz | GQA   | POPE  | CHAIR$_S$ | CHAIR$_I$ |
> |---------------|--------|-------|-------|--------|-------|--------|-------|--------|-------|-------|--------|--------|
> | CSR iter-1    | 1500.6 | 367.5 | 60.4  | 69.7   | 64.7  | 32.2   | 70.3  | 54.0   | 62.1  | 86.94 | 26.6   | 7.2    |
> | CSR iter-1 (GPT-4o)    | 1509.4 | 366.2 | 60.4  | 70.2   | 65.1  | 31.8   | 70.4  | 54.0   | 62.2  | 87.11 | 24.2   | 6.6    |
>
> ---
>
> > **Q3**: how varying $\lambda$ terms affect the score.
>
> **A3**: We evaluate the effectiveness of varying $\lambda$ on the performance. The overall training settings are the same as in Table 1, with three rounds of iteration. The results are shown in Table R3 and indicate that placing greater emphasis (i.e., $\lambda=0.9$) on visual calibration performs best, further strengthening our contribution to reward calibration.
>
> **Table R3**:  Performance w.r.t. $\lambda$
> | Method           | MME$^P$   | MME$^C$  | SEED  | LLaVA$^W$ | MMB   | MM-Vet | SQA$^I$  | VisWiz | GQA   | POPE  | CHAIR$_S$ | CHAIR$_I$ |
> |-------------------|-------|------|------|--------|------|--------|------|--------|------|------|--------|--------|
> | CSR-7B  ($\lambda$=0.1)      | 1508.6| **369.3**| 60.0 | 66.7   | 64.9 | 31.6   | 70.0 | 54.0   | 62.0 | 86.90| 40.8   | 10.2   |
> | CSR-7B  ($\lambda$=0.5)  | 1515.4| 364.5| 60.1 | 68.2   | 64.9 | 32.4   | 69.7 | 54.0   | 62.1 | 86.90| 28.2   | 6.7    |
> | CSR-7B  ($\lambda$=0.9)      | **1524.2**| 367.9 | **60.3** | **71.1**   | **65.4** | **33.9**   | **70.7** | **54.1**   | **62.3** | **87.01** | **21.0**   | **6.0**  |
>
> ---
>
> > **Q4**: Can a rough estimate of how different the computing resource would be, including the time to gather the preference data for previous methods?
>
> **A4**: In CSR, the data is constructed by the target LVLMs themselves and calibrated using the CLIP model, which eliminates the need for any human effort or costly closed-source models (e.g., GPT-4). In terms of the time required to gather the data, using LLaVA-1.5-13B as an example, it takes around 12 hours to obtain 13,000 pairs of high-quality preference data on an A6000ada GPU node. In contrast, manual data correction represented by RLHF-V [1] significantly involves human effort, as well as time and monetary costs. For POVID [2], constructing one dataset (which may not be optimal) requires approximately $300-400 in GPT-4 API costs.
>
> ---
>
> > **Q5**:  (minor) The overall framework figure (Figure 2) seems hard to understand.
>
> **A5**: We have polished Figure 2 and put it in the supplementary PDF (see Figure R2). Here is an explanation of this figure: Since CSR uses step-level reward modeling, we initially generate a few sentences (five for illustration). Each of these sentences is then used to sample five possible next sentences, resulting in a total of 25 sentences. To improve efficiency, only the top 3 sentences (indicated by green dots in the figure) and the bottom 2 sentences (indicated by red dots) are retained. The retained five sentences are then used to generate the third step sentence, and this process is repeated until the response is complete.
>
> ---
>
> **References**
>
> [1] Rlhf-v: Towards trustworthy mllms via behavior alignment from fine-grained correctional human feedback.
>
> [2] Aligning modalities in vision large language models via preference fine-tuning

---

> > ### Comment · Reviewer_myph · 2024-08-09
> >
> > Thanks for the detailed response. They have resolved my issues and I have raised the score accordingly.

---

> > > ### Author Response · Authors · 2024-08-09
> > > **Response to Reviewer myph**
> > >
> > > Dear Reviewer myph,
> > >
> > > Thank you for your response and for raising your score. We’re very pleased that our answers addressed your questions.

---

### Official Review · Reviewer_oVPD · 2024-07-13

**Soundness:** 3
**Presentation:** 3
**Contribution:** 2
**Rating:** 5
**Confidence:** 5

**Summary:**

The paper proposes a new approach to addressing the hallucination problem in Large Vision-Language Models (LVLMs). This phenomenon occurs when generated text responses appear linguistically plausible but contradict the visual input, indicating a misalignment between image and text pairs. The proposed solution, Calibrated Self-Rewarding (CSR), allows the model to self-improve by iteratively generating candidate responses, evaluating the reward for each response, and curating preference data for fine-tuning. This method incorporates visual constraints into the self-rewarding process, emphasizing visual input. Empirical results show performance improvements and reduced hallucinations across various benchmarks and tasks.

**Strengths:**

The paper addresses the critical issue of hallucination in LVLMs by introducing a new Calibrated Self-Rewarding (CSR) approach. This method is new in its use of visual constraints within the self-rewarding paradigm to enhance image-text alignment.

The empirical evaluation is thorough, encompassing twelve benchmarks and tasks. The results show improvements over existing methods, demonstrating the effectiveness of CSR.

The paper is well-organized, providing a clear explanation of the problem, proposed solution, and empirical results. The inclusion of theoretical analysis further strengthens the clarity and credibility of the work.

The proposed CSR method has implications for improving the reliability and accuracy of LVLMs, which is crucial for their application in various real-world scenarios.

**Weaknesses:**

**Technical Novelty:** The primary distinction of the proposed method from previous approaches is the introduction of a new reward score based on vision-language similarity and step-wise evaluation. While this is a meaningful contribution, the paper could benefit from a more detailed discussion on how this approach fundamentally differs from and improves upon existing methods.

**Performance Improvements:** Although CSR shows substantial improvements on average, the performance gains in some benchmarks, such as LLaVA-1.5-13B, are not very substantial. It would be beneficial to explore the reasons behind these limited improvements and suggest potential avenues for further enhancement.

**Figures:** Figures 1 and 2 lack clarity. The visual representation of preference data and the meaning of green and red circles in Figure 2, as well as the shapes and score positions in Figure 1, need to be better explained and presented more clearly.

**Questions:**

How is the improvement of approximately 7.62% computed? Please provide a detailed explanation of the calculation method used to arrive at this figure.

What is the upper bound in the performance gain of CSR? A case study showing its potential on one or several benchmarks with LLaVA-1.5 13B would be helpful, given the increasing trend shown in Figure 3.

Figure 2 is unclear. What do the green and red circles represent? What does "Option" mean? Is it possible to visualize the generated preference data in Figure 2 or elsewhere? Additionally, Figure 1 appears confusing regarding the shapes and score positions. Can these be clarified and improved for better understanding?

**Limitations:**

The authors acknowledge some limitations of their work, such as conducting three iterations of CSR. While this approach shows promise, its technical novelty compared to existing methods is somewhat limited.

Additionally, the performance improvements are not uniformly substantial across all benchmarks. The paper could benefit from a more detailed discussion of these limitations and potential strategies for addressing them in future work.

Furthermore, the potential negative societal impact of the work should be considered, particularly in terms of the ethical implications of improving LVLMs that might be used in sensitive applications. Providing constructive suggestions for mitigating any negative impacts would be valuable.

---

> ### Author Rebuttal · Authors · 2024-08-07
>
> Thank you for reviewing our paper and providing valuable feedback. We address your concerns point by point below and would appreciate knowing if our responses address them.
>
> > **Q1**: Technical Novelty: The primary distinction of the proposed method … differs from and improves upon existing methods.
>
> **A1**:
> - Compared with self-rewarding in LLM, the fundamental difference in LVLM is that using the target LVLM as both the generator and judge (as is done in LLM self-rewarding) can amplify the misalignment issues between text and image modalities. This happens because the judge itself also tends to be biased toward the contextual information, often ignoring the input image. By incorporating visual constraints, we can effectively mitigate this issue, as it encourages better alignment between text and image modalities within the judge. We believe this is the fundamental difference between our approach and LLM self-rewarding.
> - Compared with other modality alignment enhancement approaches, CSR does not rely on additional VLMs or human annotators to generate preference data, which allows it to better capture and correct the inherent preferences of the target LVLM.
>
> We believe our contributions are significant and fundamentally different from existing methods.
>
> ---
>
> > **Q2**: It would be beneficial to explore the reasons behind these limited improvements … further enhancement.
>
> **A2**: Based on Table 1, CSR performs significantly better on hallucination benchmarks that evaluate image captioning quality compared to closed-ended VQA benchmarks. This is because preference optimization may be better suited for enhancing the performance of open-ended questions. In preference optimization, the preferred and dispreferred responses in the preference data are possibly partially correct, with the preferred response simply being better than the dispreferred one. Optimizing the model with this preference data can strengthen its ability to distinguish between both responses and accurately capture more details in long-form, open-ended image captioning tasks. On the other hand, some closed-ended VQA tasks use a fixed set of response options (e.g., yes/no, multiple-choice), which rely more on the visual model's perception capability and may not be as easily improved by preference optimization compared to long-form image captioning. Similar characteristics are exhibited across POVID [1], RLHF-V [2], Human-Prefer [3], and Vlfeedback [4].
>
> ---
>
> > **Q3**: How is the improvement of approximately 7.62% computed?
>
> **A3**: As mentioned in Line 204, the improvements refer to the average percentage increase across all benchmarks when comparing CSR (7B) and LLaVA-1.5 7B. The method we used to calculate the average percentage increase is as follows:
>
> First, to calculate an average score on a 100-point scale, we adjusted the original values: MME$^P$ was divided by 16, and MME$^C$ was divided by 4, according to \# categories in MME. Additionally, since a lower CHAIR value indicates better performance, we standardized all metrics to follow a "higher is better" approach by transforming the CHAIR$_S$ and CHAIR$_I$ metrics into 100 - CHAIR$_S$ and 100 - CHAIR$_I$. Then, we calculated the average score by averaging the standardized values, which were used to compute the average percentage increase.
>
> Regarding the specific value, we apologize for a computational error in calculating the average percentage increase using the data from Table 1. The correct value is 8.23%.
>
> ---
>
> > **Q4**: What is the upper bound in the performance gain of CSR? A case study would be helpful.
>
> **A4**: We have included two case studies selected from the CSR-generated datasets in Figure R1 in the supplementary PDF. As the CSR iteration increases, we observe performance gains, the alleviation of challenging hallucinations (such as counting issues), and enhanced fine-grained perception. Combined with the qualitative improvements of CSR in various benchmarks, these additional case studies further support the effectiveness of CSR.
>
> ---
>
> > **Q5**: Figure 2 is unclear.
>
> **A5**: Since CSR uses step-level reward modeling, we initially generate a few sentences (five for illustration). Each of these sentences is then used to sample five possible next sentences, resulting in a total of 25 sentences. To improve efficiency, only the top 3 sentences (indicated by green dots in the figure) and the bottom 2 sentences (indicated by red dots) are retained. The retained five sentences are then used to generate the third step sentence, and this process is repeated until the response is complete. We have refined Figure 2 in the supplementary PDF (see Figure R2).
>
> ---
>
> > **Q6**: visualize the generated preference data
>
> **A6**: We visualized a case of preference data generated by the model itself in the CSR process in the supplementary PDF (see Figure R3).
>
> ---
>
> > **Q7**: Additionally, Figure 1 appears confusing.
>
> **A7**: Figure 1’s radar chart shows the scores of self-rewarding, LLaVA-1.5-7B, and CSR-7B across all benchmarks. A larger area in the radar chart indicates higher scores. To clarify it, we polish Figure 1 in the supplementary PDF (see Figure R4).
>
> ---
>
> > **Q8**: The potential negative societal impact of the work should be considered
>
> **A8**:  CSR depends on the quality of self-generated preference data. Although it achieves overall higher performance, there is still a possibility that the data can contain errors, which may misguide the learning process in some situations and ultimately lead to erroneous decisions in safety-critical applications like healthcare.
>
> ---
>
> **References**
>
> [1] Aligning modalities in vision large language models via preference fine-tuning.
>
> [2] Rlhf-v: Towards trustworthy mllms via behavior alignment from fine-grained correctional human feedback.
>
> [3] Aligning large multimodal models with factually augmented rlhf.
>
> [4] Silkie: Preference distillation for large visual language models.

---

> > ### Comment · Area_Chair_XWxf · 2024-08-11
> > **Please discuss with authors**
> >
> > Dear Reviewer oVPD:
> >
> > Please respond to author rebuttal and discuss with authors.
> >
> > Thanks,
> >
> > Your AC

---

### Official Review · Reviewer_GvTb · 2024-07-16

**Soundness:** 3
**Presentation:** 4
**Contribution:** 3
**Rating:** 7
**Confidence:** 4

**Summary:**

This paper proposes a new method for preference alignment with LVLMs. Specifically, the reward is computed using its own LLM (text only) and an external CLIP model. The optimization is done with DPO. This process can be iterated for several times.

**Strengths:**

S1. Preference optimization in LVLM is under explored compared to those pure language ones. This submission proposed an effective approach.

S2. Both theoretical and empirical contribution.

**Weaknesses:**

I don't find significant weaknesses. However, I do think the experiments can be improved. Therefore, I give a conservative score of borderline accept. I may consider increasing the rating if more thorough studies are conducted (see next section).

**Questions:**

Q1. How did the performance improve over each preference data generation stage?

Q2. Scaling study as in Gao et al. [1]?


[1] Gao et al., "Scaling Laws for Reward Model Overoptimization", ICML 2023

**Limitations:**

Limitations and broader impact are discussed in the appendix.

---

> ### Author Rebuttal · Authors · 2024-08-07
>
> Thank you for your valuable feedback to help us improve our paper. We detail our response below and please kindly let us know if our response addresses your concerns.
>
> > **Q1**: How did the performance improve over each preference data generation stage?
>
> **A1**: In the first round of training, our model (referred to as model-iter0) generated a batch of preference data based on the approach discussed in Section 3.2.1. Then, by employing preference optimization, we optimized model-iter0 and obtained model-iter1. Compared to model-iter0, model-iter1 exhibits better performance and generates responses that are better aligned with the image input.
> In the second round of training, similarly, we used model-iter1 to generate a batch of preference data. Due to the improved performance, this batch of data is of higher quality compared to the previous round’s preference data and is more challenging to distinguish between preferred and dispreferred data. Consequently, using this new preference data to perform preference optimization on model-iter1 yields a stronger model, i.e., model-iter2.
>
> In this way, we can perform more rounds of iteration until convergence (see empirical results in Figure 3 in the paper).
>
> In summary, because the quality of preference data can improve over different rounds and it becomes increasingly difficult for the model to distinguish preferences, performing preference optimization each round will enhance the model’s performance.
>
> ---
>
> > **Q2**: .Scaling study-Gao et al., "Scaling Laws for Reward Model Overoptimization", ICML 2023
>
> **A2**: Since CSR utilizes the model itself to establish an initial reward and then calibrates this initial reward by incorporating an image-response relevance score computed by the CLIP model, we analyzed the effect of the reward model's (CLIP model) size. We conducted experiments with different sizes of CLIP models to calibrate the initial reward. Due to rebuttal time constraints, all experiments were conducted with only one round of CSR training on LLaVA-1.5-7B. The results are reported in Table R1. The experimental results show that using larger and more powerful CLIP models as the reward model reduces the model's hallucinations and improves comprehensive benchmark performance. The results meet with our expectations, as stronger models can better align image and text responses.
>
>
> **Table R1:** Performance with respect to different sizes of CLIP models
> | CLIP models (from small to large)       | MME$^P$   | MME$^C$  | SEED  | LLaVA$^W$ | MMB   | MM-Vet | SQA$^I$  | VisWiz | GQA   | POPE  | CHAIR$_S$ | CHAIR$_I$ |
> |---------------|--------|-------|-------|--------|-------|--------|-------|--------|-------|-------|--------|--------|
> | clip-vit-base-patch16   | 1504.1 | 350.3 | 60.3  | 65.3   | 64.9  | 31.6   | 70.1  | 53.6   | 62.1  | 85.98 | 31.7   | 8.4    |
> | clip-vit-l-336px     | 1500.6 | 367.5 | 60.4  | 69.7   | 64.7  | 32.2   | 70.3  | 54.0   | 62.1  | 86.94 | 26.6   | 7.2    |
> | clip-vit-g-14   | 1511.3 | 367.1 | 60.6  | 70.6   | 65.3  | 33.0   | 70.3  | 54.3   | 62.3  | 87.02 | 26.2   | 7.1    |

---

> > ### Comment · Reviewer_GvTb · 2024-08-07
> >
> > Thanks for the response, I appreciate the additional results. My main concerns are addressed so I will raise my rating to 6. (Sorry I overlooked Fig. 3 in my initial review.)
> >
> > Following A1, is it possible to add iter-4 results (even it's negative)? Since for the 13B model it seems the performance not yet converges. Moreover, do you observe "model collapse" such that the reward keep increasing yet the benchmark score drops? If so, I think plotting the curve along the training progress could improve this submission. Following Q2&A2, I believe readers will also be interested in how reward model sizes affect the over-optimization. I'm open to raising the score again if the authors post new interesting results during the discussion period.

---

> ### Author Response · Authors · 2024-08-11
> **Response to Follow-Up Questions from Reviewer GvTb (1/2)**
>
> Thanks a lot for increasing your score, we are happy that our response addresses your concerns. For the follow up questions, we detail our response below and please kindly let us know if our response addresses your concerns.
>
>
> > **Q1**: Following A1, is it possible to add iter-4 results (even it's negative)? Since for the 13B model it seems the performance not yet converges.
>
> **A1**: We added additional iteration results for the LLaVA-7B and 13B models, which are presented in Tables R1 and R2, respectively. Due to the time constraints of the discussion period, we conducted four additional iterations for the 7B model (iter-4 through iter-7) and two additional iterations for the 13B model (iter-4 and iter-5). As shown in the "Avg score" column in both Table R1 and Table R2, the overall improvement tends to slow down or even fluctuate with each additional iteration, indicating that the performance is converging.
>
> **Table R1**: Results of additional iteration evaluation of 7B model
> | Method        | MME$^P$   | MME$^C$  | SEED  | LLaVA$^W$ | MMB   | MM-Vet | SQA$^I$  | VisWiz | GQA   | POPE  | CHAIR$_S$ | CHAIR$_I$ | **Avg score** |
> |---------------|--------|-------|-------|--------|-------|--------|-------|--------|-------|-------|--------|--------|--------|
> | CSR iter-3  | 1524.2 | 367.9 | 60.3  | 71.1   | 65.4  | 33.9   | 70.7  | 54.1   | 62.3  | 87.01 | 21.0   | 6.0    |72.09|
> | CSR iter-4  | 1524.6 | 368.8 | 60.4  | 71.0   | 65.3  | 33.9   | 70.4  | 54.0   | 62.2  | 87.05 | 19.0   | 5.9    |72.24|
> | CSR iter-5  | 1520.1 | 367.2 | 60.5  | 71.3   | 65.4  | 33.8   | 70.8  | 54.2   | 62.4  | 87.16 | 18.3   | 5.4    |72.39|
> | CSR iter-6  | 1521.3 | 365.4 | 60.4  | 70.9   | 65.3  | 34.0   | 70.7  | 54.0   | 62.4  | 87.15 | 17.9   | 5.2    |72.35|
> | CSR iter-7  | 1520.8 | 360.3 | 60.2  | 71.1   | 65.2  | 33.7   | 70.3  | 54.1   | 62.3  | 87.12 | 18.8   | 5.6    |72.06|
>
> **Table R2**: Results of additional iteration evaluation of 13B model
> | Method        | MME$^P$   | MME$^C$  | SEED  | LLaVA$^W$ | MMB   | MM-Vet | SQA$^I$  | VisWiz | GQA   | POPE  | CHAIR$_S$ | CHAIR$_I$ |**Avg score**|
> |---------------|--------|-------|-------|--------|-------|--------|-------|--------|-------|-------|--------|--------|--------|
> | CSR iter-3  | 1530.6 |303.9| 62.9| 74.7| 68.8| 37.8| 75.1| 56.8| 63.7 | 87.30 | 28.0   | 7.3    |71.95|
> | CSR iter-4  | 1530.4 | 301.4 | 63.0  | 74.2   | 68.3  | 37.3   | 75.2  | 56.6   | 63.4  | 87.20 | 27.4   | 7.4    |71.78|
> | CSR iter-5  | 1531.1 | 302.2 | 62.8  | 74.0   | 68.2  | 37.4   | 74.8  | 56.7   | 63.7  | 87.18 | 27.2   | 7.6    |71.77|
>
>
> ---
>
> > **Q2**: do you observe "model collapse" such that the reward keep increasing yet the benchmark score drops
>
> **A2**: We analyzed the relationship between the reward score and the final average performance of LLaVA-7B, with the results presented in Table R3. In the later stages of training, it can be observed that a slight increase in the reward score coincided with a slight decrease in model performance. One possible reason is that the model's performance had mostly converged, yet it continued to overfit the training data in pursuit of a higher reward score. Another explanation could be the inherent limitations of the target LVLM and the reward-calibrating model, making it increasingly challenging to accurately assign rewards to certain difficult data points. Early in training, these mis-rewarded data points may not have had a significant impact because the majority of the data was useful. However, as the model's capabilities improved, the mis-rewarded data began to affect overall performance. We believe that introducing more diverse vision-centric tools for reward calibration, along with a more thoughtful data selection mechanism, could potentially alleviate this issue. We plan to systematically explore these methods further in the future.
>
> **Table R3**: Results of rewards w.r.t. average score
> |     | iter-1 | iter-2 | iter-3 |  iter-4| iter-5| iter-6| iter-7|
> |-------------|--------|--------|--------|--------|--------|--------|--------|
> | Chosen reward          | 0.4885 | 0.5040 | 0.5052 | 0.5055| 0.5066|0.5078|0.5079|
> | Rejected reward          | 0.4551 | 0.4788 | 0.4789 |0.4794|0.4799|0.4805|0.4812|
> | Avg performance score         |66.61 | 71.02 | 71.74 | 72.09 | 72.24 | 72.39 | 72.35 | 72.06 |

---

> > ### Author Response · Authors · 2024-08-11
> > **Response to Follow-Up Questions from Reviewer GvTb (2/2)**
> >
> > > **Q3**: Following Q2&A2, I believe readers will also be interested in how reward model sizes affect the over-optimization
> >
> > **A3**: Due to the time constraints of the discussion period, we conducted experiments on LLaVA-1.5-7B using the following reward models (CLIP models): 7 iterations with clip-vit-l-336px (the model used in the paper, which is a relatively small model); and 3 iterations with clip-vit-g-14 (relatively large model). The results are reported in Tables R4 and R5. We observe that while the stronger reward model provides better feedback earlier, it also causes the target LVLM to converge more quickly.
> >
> > **Table R4**: Performance of **clip-vit-l-336px (small model)** w.r.t. CSR iterations
> >
> > | Method        | MME$^P$   | MME$^C$  | SEED  | LLaVA$^W$ | MMB   | MM-Vet | SQA$^I$  | VisWiz | GQA   | POPE  | CHAIR$_S$ | CHAIR$_I$ |**Avg score**|
> > |---------------|--------|-------|-------|--------|-------|--------|-------|--------|-------|-------|--------|--------|--------|
> > | CSR iter-1  | 1500.6| 367.5| 60.4| 69.7| 64.7| 32.2| 70.3| 54.0| 62.1  | 86.94 | 26.6   | 7.2    |71.02|
> > | CSR iter-2  | 1519.0| 368.9 |60.3| 70.4| 65.2| 33.7 |70.1 |54.0 |62.3 | 86.82 | 23.0   | 6.1    |71.74|
> > | CSR iter-3  | 1524.2 | 367.9 | 60.3  | 71.1   | 65.4  | 33.9   | 70.7  | 54.1   | 62.3  | 87.01 | 21.0   | 6.0    |72.09|
> > | CSR iter-4  | 1524.6 | 368.8 | 60.4  | 71.0   | 65.3  | 33.9   | 70.4  | 54.0   | 62.2  | 87.05 | 19.0   | 5.9    |72.24|
> > | CSR iter-5  | 1520.1 | 367.2 | 60.5  | 71.3   | 65.4  | 33.8   | 70.8  | 54.2   | 62.4  | 87.16 | 18.3   | 5.4    |72.39|
> > | CSR iter-6  | 1521.3 | 365.4 | 60.4  | 70.9   | 65.3  | 34.0   | 70.7  | 54.0   | 62.4  | 87.15 | 17.9   | 5.2    |72.35|
> > | CSR iter-7  | 1520.8 | 360.3 | 60.2  | 71.1   | 65.2  | 33.7   | 70.3  | 54.1   | 62.3  | 87.12 | 18.8   | 5.6    |72.06|
> >
> > **Table R5**: Performance of **clip-vit-l-336px  (large model)** w.r.t. CSR iterations
> >
> > | Method        | MME$^P$   | MME$^C$  | SEED  | LLaVA$^W$ | MMB   | MM-Vet | SQA$^I$  | VisWiz | GQA   | POPE  | CHAIR$_S$ | CHAIR$_I$ |**Avg score**|
> > |---------------|--------|-------|-------|--------|-------|--------|-------|--------|-------|-------|--------|--------|--------|
> > | CSR iter-1   | 1511.3 | 367.1 | 60.6  | 70.6   | 65.3  | 33.0   | 70.3  | 54.3   | 62.3  | 87.02 | 26.2   | 7.1    | 72.06|
> > | CSR iter-2  | 1519.6 | 365.9 | 60.5  | 71.9   | 65.4  | 34.0   | 70.7  | 54.1   | 62.2  | 87.21 | 20.4   | 7.0    |72.09|
> > | CSR iter-3  | 1512.1| 367.2 |60.4| 70.8| 65.2| 33.2 |70.2 |54.0 |62.4 | 87.13 | 23.9   | 7.4    |71.53|

---

> > > ### Comment · Reviewer_GvTb · 2024-08-12
> > >
> > > Thanks for the additional results! I'm further increasing my ratings and confidence.

---

> > > > ### Author Response · Authors · 2024-08-14
> > > > **Response to Reviewer GvTb**
> > > >
> > > > Thanks a lot for raising your score and your constructive comments!

---

### Author Rebuttal · Authors · 2024-08-07

We sincerely thank all reviewers for their constructive feedback. Below is a summary of the information covered in the attached PDF:

- **Figure R1**: Two cases selected from the CSR-generated datasets (Reviewer oVPD).

- **Figure R2**: Polished Figure 2 of the main paper (Reviewers oVPD, myph, 4vrA).

- **Figure R3**: A case including both self-generated preferred and dispreferred responses (Reviewer oVPD).

- **Figure R4**: Polished Figure 1 of the main paper (Reviewer oVPD).

- **Figure R5**: The average score of CSR across different iterations, with additional iterations included.

- **Figure R6 & Table A1**: We apologize for a computational error in calculating the average score of our method and LLaVA-1.5 using the data from Table 1 in the main paper. This mistake slightly affected specific values in Figure 3 and Table 2, but the improvement ratio and the overall conclusion completely remain unchanged. We have corrected this issue and updated Figure 3 in Figure R6 and Table 2 in Table A1, respectively.

For all reviewer reference, the way we **calculate the average score** is:

First, to facilitate the calculation of an average score on a 100-point scale, we adjusted the benchmarks as follows: we divided the MME$^P$ by 16 and the MME$^C$ by 4, according to the number of categories in MME. Additionally, since the CHAIR metric is better when it's lower, we unified all metrics to follow a "higher is better" approach by transforming the CHAIR$_S$ and CHAIR$_I$ metrics into 100 - CHAIR$_S$ and 100 - CHAIR$_I$, respectively. After these adjustments, we calculated the average score by summing the standardized values and dividing by the total number of datasets. We will clarify this calculation process in the revised version.

---

### Decision · Program_Chairs · 2024-09-25

**Decision:**

Accept (poster)

**Comment:**

This paper proposes a new method called Calibrated Self-Rewarding (CSR) to address hallucination issues in large vision-language models. It allows the model to self-improve by iteratively generating candidate responses, evaluating the reward for each response, and curating preference data for fine-tuning. The reward is computed using its own LLM (text only) and an external CLIP model.  Overall, all reviewers agree the approach is novel and the empirical evaluation is thorough, encompassing twelve benchmarks and tasks. However, there are some weaknesses including technical novelty is somewhat limited, and performance gains are modest on some benchmarks.
Overall, while not groundbreaking, this paper makes a solid contribution to an important problem in multimodal AI. The AC recommend weak accept.